# LEARNING MEDIUM-SENSITIVITY FUNCTIONS: A CASE STUDY ON QR CODE DECODING

## ABSTRACT

The hardness of learning a function that attains a target task relates to its input-sensitivity. For example, image classification tasks are input-insensitive as minor corruptions should not affect the classification results, whereas arithmetic and symbolic computation, the learning of which has been recently attracting interest, are highly input-sensitive as each input variable connects to the computation results. This study investigates the learning functions of medium sensitivity through learning-based Quick Response (QR) code decoding, which has both sensitivity to the change of plain texts and insensitivity to the bit flips. Our experiments reveal that Transformers can robustly decode QR codes, even beyond the theoretical error-correction limit, while remaining sensitive to single-character changes in plain texts. We demonstrate that the robust decoding ability is derived from the regularity of natural language words. Transformers trained on English-based datasets learn to exploit it. Interestingly, this generalizes to words in different languages and to random alphabetical strings. To our knowledge, this study provides the first case study of learning medium-sensitivity functions and also suggests potential applications of learning-based QR code decoding that boost classical methods in combination.

## 1 INTRODUCTION

Over a decade, deep learning has shown remarkable success particularly in learning *input-insensitive* functions that realize the target tasks. Indeed, in most standard tasks, such as image classification, object detection, document summarization, and speech recognition, a slight change in their input (e.g., image perturbation, a few typos) is supposed to have little impact on the output (e.g., classification, document summary). Consequently, for example, data augmentation such as rotation, flipping, and cropping of images has been an important technique, as it enhances the models' input-insensitivity (i.e., *robustness*), substantially increasing the performance in the tasks.

In the last few years, Transformer models (Vaswani et al., 2017) have found new applications in arithmetic and symbolic computation, where learning high-sensitivity functions attracts interest. Examples include integer and modular arithmetic (Power et al., 2022; Shen et al., 2023), symbolic integration (Lample & Charton, 2020), Lyapunov function design (Alfarano et al., 2024), Gröbner basis computation (Kera et al., 2024; 2025), and so forth (Wenger et al., 2022; Li et al., 2023a;b; Charton, 2024). In such tasks, changing a single number, coefficient, variable, or operator in an input can immediately change the output, and thus, the target functions to learn are highly input-sensitive. Although the theoretical analysis has suggested the hardness of learning such functions (Shalev-Shwartz et al., 2017; Hahn & Rofin, 2024), recent empirical and theoretical studies have shown that high-sensitivity functions can be learned through techniques such as normalization, weight decay, and autoregressive generation (Chiang & Cholak, 2022; Zhou et al., 2024; Kim & Suzuki, 2025). As such, the sensitivity of target functions plays a critical role in determining the difficulty of learning, offering new insights into the capabilities of deep learning models. However, to our knowledge, few studies have approached this perspective except for theoretical works on extremely high-sensitivity functions, particularly the parity function (Shalev-Shwartz et al., 2017; Daniely & Malach, 2020; Chiang & Cholak, 2022; Hahn & Rofin, 2024; Kim & Suzuki, 2025).

This study examines the intermediate case, i.e., learning medium-sensitivity functions, through the decoding task of Quick Response codes (QR codes; International Organization for Standardization

(2024)) using a Transformer. QR codes are designed to be tolerant (i.e., insensitive) to corruptions from image capture or physical damage, but decoding them should sensitively capture the change in the plain texts (i.e., embedded strings, such as URLs). Such characteristics place QR code decoding between the low-sensitivity case as classical tasks and the high-sensitivity case as arithmetic and symbolic tasks. QR codes have several error-correction levels and embedding capacities through which we can control their sensitivity. To the best of our knowledge, no prior studies have addressed learning-based QR code decoding. Some have tackled QR code *detection* (Chou et al., 2015; Kurniawan et al., 2019; Pu et al., 2019; Peng et al., 2020; Shindo et al., 2022; Edula et al., 2023; Zheng et al., 2023; Chen et al., 2024), which is a computer vision task and orthogonal to our scope.

Our experiments demonstrate a striking success of Transformer-based decoding under corruption. Given the typical QR-code application, we construct datasets of URLs that combine random English words and the top one million domain names from the Tranco (Pochat et al., 2018). The success rate of decoding under corruptions exceeds the theoretical limit, outperforming the standard decoding protocol. Further experiments explain this; Transformers exploit the intrinsic structure of English words (e.g., consonants and vowels roughly appear alternatingly). Nevertheless, we also demonstrate that this is not a mere memorization but generalizes to non-English words and even to random alphabetic strings.

To summarize, this study addresses a novel learning-based QR code decoding as a showcase of learning medium-sensitivity functions. We empirically reveal the following results through extensive experiments on domain name datasets:

- Transformer-based decoding empirically attains a high success rate with low corruption and maintains moderate success even when the corruption magnitude exceeds the theoretical limit of error correction, which we newly derived for analysis.

- Transformer-based QR code decoders, trained on English-rich data, generalize to other languages, and surprisingly, to random alphabetic strings as well. The former generalization suggests that Transformers acquire regularity of natural language spelling beyond English; the latter, character-level decoding, indicates that this is achieved not by memorization.

- Transformers realize a function that is sensitive to the data-codeword region of QR codes, which is essential for decoding, while remaining insensitive to other parts, including error-correction codewords. This indicates that Transformers perform error correction through a mechanism different from standard QR code readers, suggesting the benefit of combining learning-based and traditional coding-theoretic correction for robust decoding.

## 2 RELATED WORK

**Learning High-Sensitivity Functions.** Recent studies have shown that Transformers' capability of handling high-sensitivity tasks such as arithmetic and symbolic computation (Lample & Charton, 2020; Charton, 2022; Wenger et al., 2022; Li et al., 2023a;b; Charton, 2024; Alfarano et al., 2024; Kera et al., 2024; 2025). For instance, Lample & Charton (2020) demonstrated that Transformers may surpass established computational software such as Mathematica and Matlab in accuracy in symbolic integration and and ordinary differential equation solving. Although training on high-sensitivity functions was long considered theoretically difficult (Shalev-Shwartz et al., 2017; Hahn & Rofin, 2024), recent empirical work has shown that techniques such as normalization (Chiang & Cholak, 2022), weight decay (Zhou et al., 2024), and autoregressive generation (Kim & Suzuki, 2025) can successfully surmount these challenges. Prior studies have focused on functions of low or high sensitivity, and those of medium sensitivity are underexplored. In this study, we probe QR code decoding as a showcase of learning medium sensitivity functions.

**Deep Learning-based QR Code detection.** There have been many deep learning-based methods to read QR codes robustly (Chou et al., 2015; Kurniawan et al., 2019; Pu et al., 2019; Peng et al., 2020; Shindo et al., 2022; Edula et al., 2023; Zheng et al., 2023; Chen et al., 2024). Several focus on enhancing detectability by leveraging deep learning models to accurately locate QR codes within images (Chou et al., 2015; Kurniawan et al., 2019; Peng et al., 2020; Chen et al., 2024). Other use deep learning models to restore image quality and reduce blurring or low resolution to improve recognition accuracy (Pu et al., 2019; Shindo et al., 2022; Edula et al., 2023; Zheng et al., 2023).

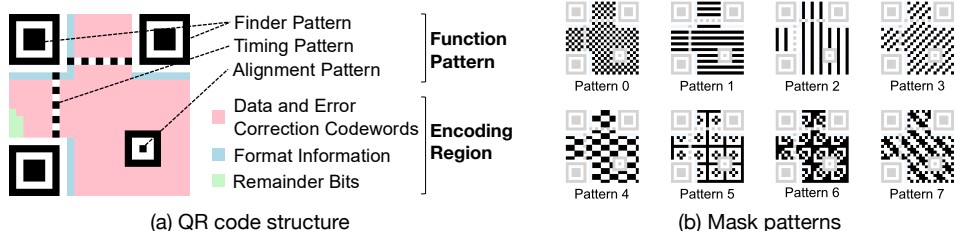

Figure 1: (a) Structure of a v2-QR code. It comprises functional patterns, which support detection, and the encoding region, which embeds the encoded representation of the plain text. (b) Mask patterns. The masking inverts the specified modules.

**Scope of This Work.** It is worth noting that prior studies on deep learning-based QR code reading focus the detection step (i.e., a computer vision task), while this study addresses decoding (i.e., a machine learning task). Besides, the QR code decoding is chosen as a showcase of medium-sensitive functions as the parity function in prior studies. Thus, the practical application is not of strong interest, while our results suggest it as a interesting future work.

## 3 QR Codes

QR codes (International Organization for Standardization, 2024) are two-dimensional matrix codes developed by Denso Wave in 1994 and designed for high-speed information reading. QR codes enable high-speed, accurate transmission of textual information and are used worldwide for various applications, including website access, electronic payments, and airline ticketing.

Figure 1(a) illustrates the basic structure of QR codes. A QR code consists of black and white units (called *modules*). Some of the modules form *function patterns*, which are used at the detection stage by cameras. The rest of the part corresponds to *encoding region*, which encodes the data and error correction bits. The encoding region splits into three components.

**Data and Error-Correction Codewords.** The plain text is encoded with the Reed–Solomon code and organized into 8-bit units called *codewords*. The data codewords represent the plain text itself, while the error-correction codewords contain the redundant information required for error correction. QR codes support four error correction levels (i.e., L, M, Q, H in ascending order). As the level increases, the ratio of error-correction codewords to the total number of codewords also increases.

**Format Information.** It encodes the error-correction level and mask pattern identifier with a BCH code. The resulting bit sequence is duplicated; one to the top-left finder pattern, and another next to the bottom-left and top-right finder patterns in the split.

**Remainder Bits.** These bits are the "leftovers," resulting from the codeword placement. They contain no actual data or error-correction information and are ignored during decoding.

Raw QR codes can have an imbalanced distribution of black and white modules. To mitigate this, masking is performed in the final encoding step. There are eight mask patterns (Figure 1(b)), and the most suitable one is selected according to a scoring rule that penalizes poor arrangements, such as long runs of the same color or layouts that hinder detection.

QR codes have 40 versions, from Version 1 to Version 40, according to their size. The higher version has more modules, implying higher data capacity. For example, Version 1 has 21x21 modules, Version 2 has 25x25, and Version 3 has 29x29. These versions are widely used in product packaging and promotional materials. Hereinafter, for example, we denote by (v3, L)-QR codes the codes in Version 3 with error correction level L.

## 4 Success Rate Analysis of Error Correction

We will later evaluate the robustness of QR code decoding by Transformers. To this end, we here derive the error-correction success rate of a standard QR code reading with $n$-bit errors. Below, we

assume that bit flips occur uniformly at random in the encoding region, as those in function patterns only affect detection, not decoding.

Recall that the encoding region consists of the data and error-correction codewords, the format information, and the remainder bits, and the remainder bits are not used in the decoding. Let $N$ denote the total number of bits in the encoding region, and let $N_{\mathrm{d}}$, $N_{\mathrm{f}}$ and $N_{\mathrm{r}}$ be the numbers of bits in the data and error-correction codewords and the format information, remainder bits, respectively. Accordingly, the numbers of erroneous bits are denoted by $p$, $q$, and $n - p - q$, respectively. Then, the success rate of error correction in the encoding region is given as follows.

**Theorem 1** (Success Rate of Error Correction in the Encoding Region). *Let $n$ be the total number of bit errors in the encoding region. The success rate $P_{\mathrm{success}}(n)$ of error correction is then:*

$$P_{\mathrm{success}}(n) = \frac{1}{\binom{N}{n}} \sum_{p=0}^{n} \sum_{q=0}^{n-p} W(p,q) P_{\mathrm{d}}(p) P_{\mathrm{f}}(q), \tag{1}$$

*where $P_{\mathrm{d}}$ and $P_{\mathrm{f}}$ respectively denote the success rate of error correction in the data and error-correction codewords with $p$-bit errors and that in format information with $q$-bit errors, and*

$$W(p,q) = \binom{N_{\mathrm{d}}}{p} \binom{N_{\mathrm{f}}}{q} \binom{N_{\mathrm{r}}}{n - p - q} \tag{2}$$

*is the probability that $(n - p - q)$-bit errors fall into the remainder bits.*

In the following, we provide an overview of the derivation of $P_{\mathrm{d}}(p)$ and $P_{\mathrm{f}}(q)$ in Equation (1). All the proofs can be found in Appendix A.

$P_{\mathrm{d}}(p)$ **- Success Rate in Data and Error-Correction Codewords.** For the data and error-correction codewords, let $M = N_{\mathrm{d}}/8$ denote the total number of codewords, and let $M_{\mathrm{ecc}}$ be the number of error-correction codewords, which depends on the error correction level. According to the properties of Reed–Solomon codes, the maximum number of correctable codewords $t = \lfloor M_{\mathrm{ecc}}/2 \rfloor$. Then, the success rate of error correction for the data and error-correction codewords can be determined as follows:

**Theorem 2** (Success Rate of Error Correction in Data and Error-Correction Codewords). *Let $p$ be the number of erroneous bits within the data and error-correction codewords. Then, the success rate $P_{\mathrm{d}}(p)$ of error correction is given by*

$$P_{\mathrm{d}}(p) = \frac{1}{\binom{N_{\mathrm{d}}}{p}} \sum_{k=\lceil p/8 \rceil}^{t} \binom{M}{k} \sum_{j=0}^{k} (-1)^j \binom{k}{j} \binom{8(k-j)}{p}. \tag{3}$$

$P_{\mathrm{f}}(q)$ **- Success Rate in Format Information.** The format information is encoded into 15 bits using a $(15, 5)$-BCH code, and the resulting bit sequence is duplicated so that two identical copies are placed in the QR code. If error correction works in either one of the two, the format information can be correctly read. The $(15, 5)$-BCH code can correct up to 3-bit errors. Under these conditions, the success rate of error correction for format information is given below:

**Theorem 3** (Success Rate of Error Correction in Format Information). *Let $q$ denote the number of erroneous bits in the format information, and let $i$ and $j$ represent the number of bit errors in each of the two respective instances. Then, the success rate $P_{\mathrm{f}}$ of error correction is given by*

$$P_{\mathrm{f}}(q) = \frac{\left| \{(i,j) \in \mathbb{Z}_{\geq 0}^2 \mid i + j = q, \ \min(i,j) \leq 3\} \right|}{q + 1}. \tag{4}$$

As will be shown in Figure 2(a), the theoretical success rate $P_{\mathrm{success}}(n)$ tightly aligns with the empirical result by `pyzbar`. To the best of our knowledge, this study is the first to derive the theoretical success rate.

## 5 LEARNING TO DECODE QR CODES

This section presents the evaluation results of the Transformer's QR code decoding success rate and robustness. We assume that QR code detection has been successfully performed and focus solely on the decoding phase, namely, retrieving the plain texts from input QR codes in the bit string format.

Table 2: Mask-pattern-wise success rate (%). A model trained on a dataset where mask patterns are selected with the scoring rule as in a practical scenario. The success rates are heavily biased to the imbalance proportion (%) of mask pattern types in the training dataset.

| Mask Pattern | 0 | 1 | 2 | 3 | 4 | 5 | 6 | 7 | Average |
|---|---|---|---|---|---|---|---|---|---|
| Success Rate | 59.6 | 92.8 | 49.1 | 64.2 | 90.9 | 50.7 | 68.8 | 54.2 | 68.3 |
| Proportion | 1.3 | 60.6 | 1.7 | 2.7 | 29.1 | 1.2 | 2.4 | 1.1 | - |

## 5.1 SETUP

**Dataset.** We constructed several URL datasets using Tranco Top 1 Million (Pochat et al., 2018) (as of November 27, 2024), a publicly available ranking of popular domain names (cf.Appendix G). The plain texts to embed were selected as a random subset of the ranking and encoded to QR codes using the Segno library[1]. QR codes were then linearized into bit strings in the order of Table 1(d). The variation of datasets comes from the combinations of QR code versions, error correction levels, and mask patterns. Each dataset contains 500,000 samples for training and 1,000 samples for evaluation.

Table 1: The success rates for different input orders in (v3, L)-QR codes. The order (d) led to the highest success rate.

| | (a) | (b) | (c) | (d) |
|---|---|---|---|---|
| Order |  |  |  |  |
| Succ. R. | 93.3 | 90.2 | 93.9 | **95.5** |

**Model and Training.** We adopted a standard architecture (Vaswani et al., 2017) (six encoder and six decoder layers, eight attention heads) and conventional training settings (AdamW optimizer (Loshchilov & Hutter, 2019) with a linearly decaying learning rate starting from $10^{-4}$). Following (Lewis et al., 2020), the input embeddings and the output projection layer shared weights, and a learnable positional embedding was used. The batch size was set to 16, and training was conducted for 10 epochs. We trained a Transformer on each dataset (i.e., a combination of QR-code version, error correction level, and mask pattern) independently.

**Evaluation Metrics and Baselines.** We measure the success rate of decoding QR codes, i.e., retrieving the embedded plain texts with and without corruptions in bit strings. Two types of corruptions are considered here: flip errors and burst errors. The former is random, independent bit flips, while the latter is bit flipping of random 3x3 square regions. The bit flips are only applied to the encoding region, and not to functional patterns, as the latter are used only for detection. We adopt `pyzbar` as a baseline, a Python wrapper of ZBar QR code[2], a widely used open-source library. Equation (1) also serves as a theoretical baseline.

**Preliminary Experiments.** The aforementioned setup comes from three preliminary observations.

1. Training was unsuccessful when the mask pattern of QR codes in the training set was selected by the standard scoring rule. This is because of the imbalanced proportion of mask patterns, see Table 2 and Appendix C.

2. It was easy to train a near-perfect classifier of the mask pattern, the accuracy of which was almost 100 % (cf. Appendix D).

3. The order of QR-code bit string affected the success rate of decoding, see Table 1. The order (d) is more successful. This is likely because it aligns more closely with the bit ordering of QR codes (cf. Appendix E).

The first two observations suggest a two-step pipeline of QR code decoding, first classifying the mask pattern of input QR codes and then sending them to Transformers specialized to each mask pattern. This allows us to focus on training Transformers on QR codes with a fixed mask pattern. From the last observation, we adopted the order (d) in our main experiments.

---

[1] https://github.com/heuer/segno
[2] https://github.com/NaturalHistoryMuseum/pyzbar

Table 3: Success rate (%) of Transformers for each mask pattern. The average decoding success rates are over 93 % for all QR code versions, suggesting Transformers can learn QR-code decoding.

| Mask Pattern | 0 | 1 | 2 | 3 | 4 | 5 | 6 | 7 | Average |
|---|---|---|---|---|---|---|---|---|---|
| Version 1 | 99.0 | 97.9 | 99.2 | 97.8 | 98.6 | 98.5 | 97.5 | 97.8 | 98.3 |
| Version 2 | 96.0 | 95.5 | 96.3 | 95.0 | 92.1 | 95.0 | 94.5 | 94.3 | 94.8 |
| Version 3 | 95.5 | 95.5 | 95.6 | 93.6 | 95.1 | 91.6 | 93.4 | 90.8 | 93.9 |

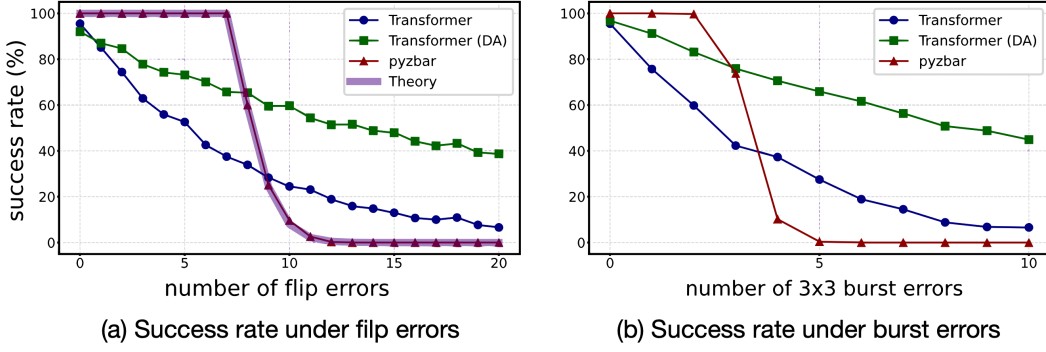

(a) Success rate under filp errors      (b) Success rate under burst errors

Figure 2: Success rate of decoding (v3, L)-QR code (mask pattern 0) with bit errors. (a) flip errors, and (b) burst errors. When the level of corruption is high, Transformers decoded with a greater success rate than the theoretical limit (*Theory*; Equation (1)). Data augmentation leads to substantial improvements (*Transformer (DA)*).

## 5.2 SUCCESS RATE AND ROBUSTNESS IN DECODING

The results show that Transformers can learn accurate QR coding and outperform the standard protocol `pyzbar` when severe corruptions are applied.

**Clean Success Rate.** Table 3 shows the decoding success rate of Transformers for (v1, L)-, (v2, L)-, and (v3, L)-QR codes with different mask patterns. Across all versions and mask patterns, the success rate exceeds 93 %, suggesting that the QR-code decoding process, a medium-sensitivity function, is learnable by Transformers.

**Robustness.** Figure 2 illustrates the evaluation results for (v3, L)-QR codes (mask pattern 0) with two types of corruptions. In both cases, `pyzbar` attained perfect decoding when the error magnitude is moderate. However, its success rate starts sharply degrading when the number of bit flips exceeds $n = 9$, while, surprisingly, that of Transformer-based decoding remains moderate. As can be seen in Figure 2(a), our theoretical result Equation (1) accurately predicts the success rate degradation of `pyzbar`. Figure 7 shows several examples of QR codes with severe corruptions where `pyzbar` fails but the Transformer succeeds. Such robust decoding by Transformers may be attributed to the fact that domain names are typically based on English words, which have certain (learnable) regularities, i.e., statistical patterns in vowels, consonants, and their combinations. Note that the plain texts in training and evaluation sets had no overlap, so the robustness is not a simple artifact of memorization. The next section will further investigate this.

**Data Augmentation.** *Transformer (DA)* in Figure 2 shows significant success rate improvement by data augmentation. Here, Transformers were trained on corrupted bit strings. The bit-flip data augmentation applies $n$-bit flips, where $n \in \{1, \ldots, 20\}$ was selected randomly and uniformly. The burst data augmentation instead applies $n$ 3x3 burst bit flips with $n \in \{1, \ldots, 10\}$. For future work, it may be interesting to investigate whether other standard data augmentation and training techniques improve the decoding ability.

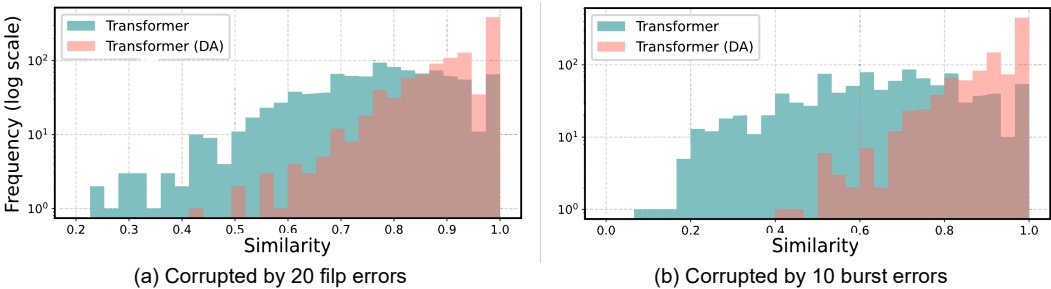

(a) Corrupted by 20 filp errors                    (b) Corrupted by 10 burst errors

Figure 3: Distributions of similarity scores for the generated strings vs. ground-truth plain texts with the input bit strings subjected to (a) 20 flip errors and (b) 10 burst errors.

Table 4: Examples of strings and their similarity scores.

| Strings | Transformer | Transformey | Transforcel | Transfuimur | Tranyfjjber |
|---|---|---|---|---|---|
| Similarity | 1.00 | 0.91 | 0.82 | 0.73 | 0.64 |
| pcavsfzumec | buagyfirchr | Trpbgfbildv | xwbiliorblq | bpqzcvprwvh | bxxggbggpjx |
| 0.45 | 0.36 | 0.27 | 0.18 | 0.09 | 0.00 |

## 5.3 FAILURE CASES

We show that Transformer outputs are reasonable even in the failure cases. We compare the reconstructed texts from Transformers and ground-truth plain texts embedded in the input QR codes using the following similarity score based on the Levenshtein distance (Levenshtein, 1966):

$$\text{Similarity}(x, y) = 1 - \frac{\text{Levenshtein}(x, y)}{\max(|x|, |y|)}, \tag{5}$$

where $x, y$ are two strings being compared. Table 4 presents several strings and the associated similarity scores.

Figure 3 illustrates the distributions of the similarity scores of the reconstructed texts and the ground-truth plain texts for severely corrupted input bit strings (i.e., 20 flip errors or 10 burst errors). One can see that similarity scores are reasonably high for most unsuccessful cases. Without data augmentation, the peaks are found around 0.7–0.8, and data augmentation significantly improves the similarity scores.

## 6 ANALYSIS FROM LINGUISTIC STRUCTURE PERSPECTIVES

This section investigates why the Transformer surpasses the theoretical error-correction limit in the previous experiment. The datasets in the previous sections were constructed from Tranco Top-1 Million, where English words are dominant. Thus, Transformers may exploit the statistical alphabetic patterns of English words and/or simply remember frequent prefixes and suffixes. We will now examine this using synthetic datasets in various structures.

## 6.1 SETUP

We construct new evaluation sets for analysis. Each set consists of 5,000 (v3, L)-QR codes (mask pattern 0). All other setups are the same as those in Section 5.1. To imitate domain names from Tranco in the previous experiment, we used the domain-name format "{$word_1$}{$word_2$}.{$tld$}," where $word_1$ and $word_2$ are concatenated to form the second-level domain (SLD), and $tld$ denotes the top-level domain (TLD), which is randomly selected from typical ones, such as "com," "org," or "co." The new evaluation sets can be classified into three categories upon the structure of the sampled words (i.e., $word_1$ and $word_2$).

Table 5: Success rate (%) across evaluation sets. For structured datasets (English, German, Swahili), the Transformer shows a high success rate. On the other hand, for unstructured datasets (Shuffle, Random-Alphabet), which lack such structure, the success rate is lower. Furthermore, for datasets with corruptions (Misspelled, Leetspeak), which retain word structures but include corruptions, the Transformer achieves the lowest success rates.

| Category | Structured | | | Unstructured | | Structured with Corruptions | |
|---|---|---|---|---|---|---|---|
| Dataset | Eng. | Ger. | Swa. | Shuffle | Rand.-Alph. | Misspelled | Leetspeak |
| Success Rate | 99.2 | 97.1 | 96.6 | 95.3 | 94.4 | 88.1 | 72.5 |

**Structured.** Three evaluation sets, *English, German, Swahili*, were constructed for each language. The words $word_1$ and $word_2$ were set to random words in the corresponding language. See Appendix H for details. The datasets in this category have the word regularity of the associated languages.

**Unstructured.** This category does not have any linguistic regularities. *Shuffle*: randomly character-level permutation of English words. *Random-Alphabet*: random sequences of alphabets.

**Structured with Corruptions.** Datasets composed of corrupted English words. *Misspelled*: A character in each word is randomly replaced with an incorrect alphabet. *Leetspeak*: A character in each word is replaced by a visually similar digit (e.g., d0g from dog).

In total, we have seven evaluation sets with three categories, and we evaluate on them a Transformer from the previous experiment, which was trained on (v3, L)-QR codes (mask pattern 0) that embed domain names from the Tranco dataset.

## 6.2 Decoding Success Rate by Word-Structure Level

Table 5 presents that the success rate varies between the dataset categories.

First, the three highest success rates are found in **Structured**, *English*, *German*, and *Swahili* in descending order. This indicates generalization to non-English words, and thus, the QR-code decoding capability of the Transformer model was not achieved by memorization. The Transformer has learned a generalizable regularity of natural languages from English words in the domain names from the Tranco dataset. The subtler success rate drop in from *English* to *German*, compared with that from *English* to *Swahili*, reflects the linguistic distance of these languages from English.

Another important and surprising observation is that the success rates in **Unstructured** are also high. Although the Transformer only observed domain names, which are mostly structured, during training, it can decode QR codes that embed random strings. This strongly indicates that the Transformer has learned QR code decoding, although not perfectly.

Nevertheless, the regularity of natural language words is exploited by Transformers for decoding QR codes robustly, as the gap of success rates between *English*, *German*, *Swahili* suggested. The last category **Structured with Corruptions** provides clearer evidence. Here, the success rate drops from that of *English* are noticeable, although retaining moderate success rates. As Table 6 shows, the Transformer performed "spell-check," leading to decoded texts with proper English words.

Table 6: Failure decoding on *Misspelled*. The Transformer "spell-checked" words.

| Ground Truth | Prediction |
|---|---|
| acrfreedo**z**.io | acrfreedo**m**.io |
| domai**j**protoc**j**l.me | domai**n**protoc**o**l.me |
| fiameupstre**r**m.site | fiameupstre**a**m.site |
| gho**c**twnreless.fr | gho**s**twnreless.fr |
| ordbrow**q**er.cc | ordbrow**s**er.cc |

To summarize, this section demonstrated that the accurate and robust QR-code decoding by Transformers observed in Section 5 is not an artifact of memorization. The evaluation on the controlled evaluation sets shows that Transformers exploit word structure for generalization and robustness while remaining reasonably sensitive to single-character changes. Namely, Transformers realize a function of medium-, or inhomogeneous-, sensitivity. Further discussion on the robust decoding can be found in Appendix K.

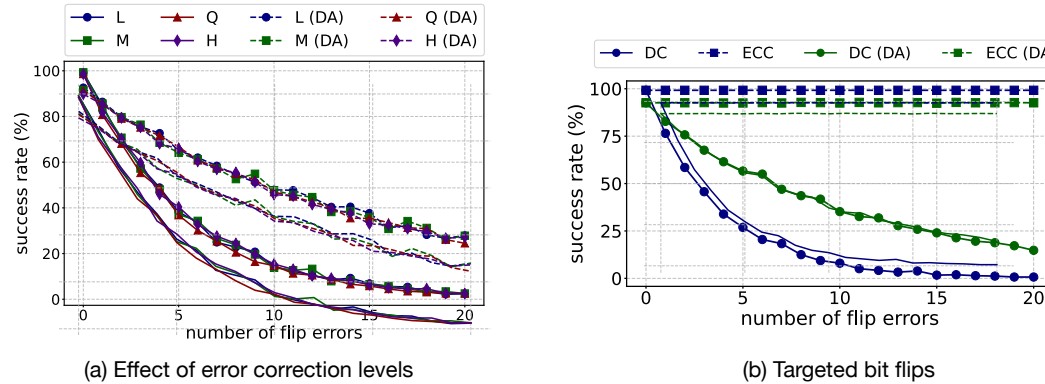

(a) Effect of error correction levels          (b) Targeted bit flips

Figure 4: (a) Success rates (%) of decoding v2-QR codes with different error correction levels under flip errors. *(DA)* indicates training with data augmentation. Success rate curves are similar across all levels. (b) Success rates (%) when flip errors in data codewords (*DC*) or in error-correction codewords (*ECC*) of (v2, L)-QR codes. While errors in the error-correction codewords have little impact, those in the data codewords significantly degrade the success rate.

## 7   TRANSFORMER'S ATTENTION ON QR CODES

We have shown that Transformers use the regularity of natural language words to robustly decode QR codes. This is particular to the learning-based approach. The standard decoding protocol instead uses error-correction codewords. This section presents that Transformers do not use such codewords, suggesting that the robustness realized by Transformers and the standard protocol is complementary.

**Setup.** Recall that QR codes support four levels of error correction (i.e., L, M, Q, and H in ascending order of strength). The level indicates a larger proportion of error-correction codewords to the total number of codewords, leading to a more powerful error correction ability. We trained Transformers for the v2-QR code decoding task with each error-correction level. All training and evaluation settings, aside from the error correction level of QR codes, were identical to those in Section 5.1. The plain texts to construct the training and test sets were all identical across error-correction levels.

**Results.** Figure 4(a) shows the success rates of decoding across the four error-correction levels under bit flip errors. While the data augmentation (*DA*) at the training stage improves the success rate, no evident difference can be observed across error correction levels. This suggests that Transformers do not exploit error-correction codewords. Figure 4(b) supports this. When the bit flips were only applied to data codewords (*DC*), the success rate decreased; however, when applied to error-correction codewords (*ECC*), the success rate remained unchanged.

To summarize, Transformers and the standard protocol do not utilize the error correction bits for QR code decoding. This indicates that their error correction is orthogonal to the coding-theoretic approach, suggesting that combining the two could lead to even more robust QR code decoding.

## 8   CONCLUSION

This study tackled the QR code decoding task as an example of a medium-sensitivity function. We demonstrated that Transformers achieve a high success rate under low corruption and maintain moderate decoding success rates even when the corruption exceeds the theoretical error-correction limit, which we newly derived. Moreover, we demonstrated that the Transformer exploits word structure for generalization and robustness while remaining highly sensitive to single-character changes. The presented robustness to corruptions and sensitivity to character changes suggest that Transformers acquired functions of medium, or inhomogeneous, sensitivity. While our findings shed light on the Transformer's fundamental behavior, pursuing a robust QR-code reader with Transformers by introducing a vision module and efficient architecture is also an interesting future work in applications.

**Reproducibility Statement and the Use of LLMs.** The setup details of experiments are described at the beginning of each experiment in Sections 5 to 7 and Appendix H. The source code used for experiments is provided as supplemental material and will be publicly available after clean-up. The LLMs were used for assistance purposes only. We used them to improve our writing and speed up coding. No essential contributions were made by the LLMs.

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

## A    SUCCESS RATE OF ERROR CORRECTION IN THE ENCODING REGION

Here, we derive the success rate of error correction in QR codes with $n$-bit errors. We assume that a bit flip occurs uniformly at random in the encoding region. The encoding region consists of the data and error-correction codewords, the format information, and the remainder bits. Note that the remainder bits are not used in the decoding; thus, any errors in those bits do not affect the result.

Let $N$ denote the total number of bits in the encoding region, and let $N_{\mathrm{d}}$, $N_{\mathrm{f}}$ and $N_{\mathrm{r}}$ be the numbers of bits in the data and error-correction codewords and the format information, remainder bits, respectively. Accordingly, the numbers of erroneous bits are denoted by $p$, $q$, and $n - p - q$, respectively. Then, the success rate of error correction is given by the following.

**Theorem 1** (Success Rate of Error Correction in the Encoding Region). *Let $n$ be the total number of bit errors in the encoding region. The success rate $P_{\mathrm{success}}(n)$ of error correction is then:*

$$P_{\mathrm{success}}(n) = \frac{1}{\binom{N}{n}} \sum_{p=0}^{n} \sum_{q=0}^{n-p} W(p, q) P_{\mathrm{d}}(p) P_{\mathrm{f}}(q), \tag{1}$$

*where $P_{\mathrm{d}}$ and $P_{\mathrm{f}}$ respectively denote the success rate of error correction in the data and error-correction codewords with $p$-bit errors and that in format information with $q$-bit errors, and*

$$W(p, q) = \binom{N_{\mathrm{d}}}{p} \binom{N_{\mathrm{f}}}{q} \binom{N_{\mathrm{r}}}{n - p - q} \tag{2}$$

*is the probability that $(n - p - q)$-bit errors fall into the remainder bits.*

In the following, we provide an overview of the derivation of $P_{\mathrm{d}}(p)$ and $P_{\mathrm{f}}(q)$.

$P_{\mathrm{d}}(p)$ **- Success Rate in Data and Error-Correction Codewords.** For the data and error-correction codewords, let $M$ denote the total number of codewords, and let $M_{\mathrm{ecc}}$ represent the number of error-correction codewords, which depends on the error correction level. Here, $M = N_{\mathrm{d}}/8$. According to the properties of Reed–Solomon codes, the maximum number of correctable codewords, denoted by $t$, is determined by $t = \lfloor M_{\mathrm{ecc}}/2 \rfloor$. Under these conditions, the probability of successful error correction for the data and error-correction codewords can be determined as follows:

**Theorem 2** (Success Rate of Error Correction in Data and Error-Correction Codewords). *Let $p$ be the number of erroneous bits within the data and error-correction codewords. Then, the success rate $P_{\mathrm{d}}(p)$ of error correction is given by*

$$P_{\mathrm{d}}(p) = \frac{1}{\binom{N_{\mathrm{d}}}{p}} \sum_{k=\lceil p/8 \rceil}^{t} \binom{M}{k} \sum_{j=0}^{k} (-1)^j \binom{k}{j} \binom{8(k - j)}{p}. \tag{3}$$

*Proof.* Let $K$ denote the number of codewords that contain at least 1-bit error. Since the Reed–Solomon decoder can correct up to $t$ codewords with errors, the Success Rate is given by:

$$P_{\mathrm{d}}(p) = P(K \leq t)$$

$$= \sum_{k=\lceil \frac{p}{8} \rceil}^{t} P(K = k) \tag{6}$$

$$= \frac{1}{\binom{N_{\mathrm{d}}}{n}} \sum_{k=\lceil \frac{p}{8} \rceil}^{t} |\{K = k\}|.$$

Let $S_k(p)$ denote the number of ways in which exactly $k$ distinct codewords each contain at least 1-bit error, given that there are $p$ bit errors in total. Then, under the condition that all $k$ selected codewords include at least one erroneous bit, the value of $S_k(p)$ is given by

$$|\{K = k\}| = \binom{M}{k} S_k(p). \tag{7}$$

Let $U$ be the set of all ways to choose $n$ error-bit positions out of the $8k$ bits of the $k$ selected codewords (so $|U| = \binom{8k}{p}$). Define $A_i \subseteq U$ as the subset of error assignments in which the $i$-th codeword contains no bit errors. Then, the number of assignments in which every codeword contains at least 1-bit error is given by:

$$S_k(p) = \left| \bigcap_{i=1}^{k} A_i^c \right|$$

$$= |U| - \left| \bigcup_{i=1}^{k} A_i \right|. \tag{8}$$

To evaluate the number of assignments in which every codeword contains at least one erroneous bit, we apply the principle of inclusion-exclusion, as formalized in Lemma 1. Specifically, applying it to the sets $A_1, A_2, \ldots, A_k \subseteq U$, we obtain:

$$\left| \bigcup_{i=1}^{k} A_i \right| = \sum_{j=1}^{k} (-1)^{j-1} \sum_{L \subseteq [k], |L|=j} \left| \bigcap_{l \in L} A_l \right|. \tag{9}$$

The term $\left| \bigcap_{l \in L} A_l \right|$ denotes the number of cases in which $j$ codewords, selected from the $k$ total codewords, are all error-free; thus,

$$\sum_{L \subseteq [k], |L|=j} \left| \bigcap_{l \in L} A_l \right| = \binom{k}{j} \binom{8(k-j)}{p}. \tag{10}$$

From Equations (8) to (10), it follows that

$$S_k(p) = |U| - \left| \bigcup_{i=1}^{k} A_i \right|$$

$$= \binom{8k}{p} - \sum_{j=1}^{k} (-1)^{j-1} \binom{k}{j} \binom{8(k-j)}{p}$$

$$= \binom{8k}{p} + \sum_{j=1}^{k} (-1)^{j} \binom{k}{j} \binom{8(k-j)}{p}$$

$$= \sum_{j=0}^{k} (-1)^{j} \binom{k}{j} \binom{8(k-j)}{p}. \tag{11}$$

Therefore, from Equations (6), (7) and (11), we obtain

$$P_{\mathrm{d}}(p) = \frac{1}{\binom{N_{\mathrm{d}}}{p}} \sum_{k=\lceil \frac{p}{8} \rceil}^{t} |K = k|$$

$$= \frac{1}{\binom{N_{\mathrm{d}}}{p}} \sum_{k=\lceil \frac{p}{8} \rceil}^{t} \binom{M}{k} \sum_{j=0}^{k} (-1)^{j} \binom{k}{j} \binom{8(k-j)}{p}. \tag{12}$$

$\square$

$P_{\mathrm{f}}(q)$ **- Success Rate in Format Information.** The format information is encoded into 15 bits using a $(15, 5)$-BCH code and placed in two separate locations in the QR code. If error correction works in either one of the two, the format information can be correctly read. The $(15, 5)$-BCH code can correct up to 3-bit errors. Under these conditions, the probability that the format information is correctly recovered is given below:

**Theorem 3** (Success Rate of Error Correction in Format Information). *Let $q$ denote the number of erroneous bits in the format information, and let $i$ and $j$ represent the number of bit errors in each of the two respective instances. Then, the success rate $P_{\mathrm{f}}$ of error correction is given by*

$$P_{\mathrm{f}}(q) = \frac{\left| \{ (i,j) \in \mathbb{Z}_{\geq 0}^2 \mid i+j = q, \, \min(i,j) \leq 3 \} \right|}{q+1}. \tag{4}$$

*Proof.* When exactly $q$ bit-errors occur in the format information, there are $q + 1$ ways to distribute those errors across the two $(15, 5)$ BCH codeword blocks. Decoding of the format information succeeds if at least one of the two blocks contains at most 3-bit errors. Labeling by $i$ and $j$ the numbers of errors in the first and second blocks respectively, the number of allocations that lead to successful decoding can be written as $|\{(i, j) \in \mathbb{Z}_{\geq 0}^2 \mid i + j = q, \ \min(i, j) \leq 3\}|$. Hence,

$$P_{\mathrm{f}}(q) = \frac{|\{(i, j) \in \mathbb{Z}_{\geq 0}^2 \mid i + j = q, \ \min(i, j) \leq 3\}|}{q + 1}. \tag{13}$$

$\square$

## B  PRINCIPLE OF INCLUSION–EXCLUSION

**Lemma 1** (Principle of Inclusion–Exclusion). *Let $A_1, A_2, \ldots, A_k$ be subsets of a finite set $U$. Then, the cardinality of their union is given by:*

$$\left| \bigcup_{i=1}^k A_i \right| = \sum_{j=1}^k (-1)^{j-1} \sum_{L \subseteq [k], |L|=j} \left| \bigcap_{l \in L} A_l \right|, \tag{14}$$

*where $[k] = \{1, 2, \ldots, k\}$.*

*Proof.* We prove the identity by counting how many times each element $x \in U$ is counted on the right-hand side. Fix an element $x \in U$, and suppose that $x$ belongs to exactly $m$ of the sets $A_1, A_2, \ldots, A_k$. Without loss of generality, assume $x \in A_1, A_2, \ldots, A_m$, and $x \notin A_{m+1}, \ldots, A_k$. On the right-hand side, $x$ contributes to each term of the form $\left| \bigcap_{l \in L} A_l \right|$, where $L \subseteq [k]$ and $x \in A_l$ for all $l \in L$. Since $x$ belongs to exactly $m$ of the $A_i$, the total number of such contributing terms is:

$$(-1)^0 \binom{m}{1} + (-1)^1 \binom{m}{2} + \cdots + (-1)^{m-1} \binom{m}{m}$$
$$= \sum_{j=1}^m (-1)^{j-1} \binom{m}{j}. \tag{15}$$

This sum can be evaluated using the binomial theorem:

$$0 = (1 - 1)^m$$
$$= \sum_{j=0}^m (-1)^j \binom{m}{j}$$
$$= 1 - \sum_{j=1}^m (-1)^{j-1} \binom{m}{j}. \tag{16}$$

From Equation (16), it follows that

$$\sum_{j=1}^m (-1)^{j-1} \binom{m}{j} = 1. \tag{17}$$

Therefore, every $x \in \bigcup_{i=1}^k A_i$ contributes exactly once to the right-hand side. Elements not in the union contribute nothing. Hence, the right-hand side counts exactly the number of elements in the union. $\square$

## C  TRAINING ON QR CODES WITH MASK PATTERNS IN MIXTURE

In practice, mask patterns of QR codes are selected by the scoring criteria (International Organization for Standardization, 2024). In this section, we construct the dataset by encoding v3-QR codes from internet-domain inputs with mask patterns chosen automatically by the scoring rules and subsequently train the Transformer. For evaluation, we prepare 1,000 samples for each of the eight

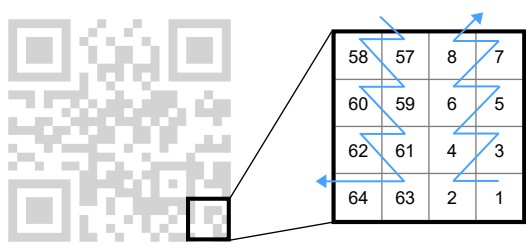

Figure 5: Data arrangement in v2-QR Code, placed in vertical zigzags within two-column bands while avoiding function patterns.

mask patterns, yielding 8,000 samples. The experimental configuration follows that of Section 5.1. As presented in Table 2, the Transformer attains an average decoding success rate of 68.3 % when mask patterns are heterogeneous. Although only patterns 1 and 4 achieve decoding accuracies above 90 %, the remaining patterns fall below the 3 % threshold. We attribute this disparity to the biased proportion of mask patterns within the training dataset. Table 2 depicts this proportion, revealing that most of the 500,000 training samples correspond to patterns 1 and 4. This pronounced imbalance results in an insufficient representation of the other patterns, impairing generalization and reducing inference performance.

## D    MASK PATTERN CLASSIFICATION

To ascertain whether a Transformer can discriminate among mask patterns, we trained and evaluated the model using a dataset encompassing all eight mask patterns. For training, we curated 800,000 popular internet domains from the Tranco list and synthesized the dataset such that each mask pattern was represented by 100,000 samples. For evaluation, we collected 8,000 samples from internet domains distinct from the training set, ensuring each mask pattern comprised 1,000 samples. The architecture and training protocol were identical to those described in Section 5.1. We set the batch size to eight and trained the model for one epoch. The evaluation revealed that the Transformer achieved a classification accuracy of 100 %. This result confirms that mask patterns can be classified with perfect accuracy; consequently, in our proposed method, we assume known mask patterns and train separate models for each pattern.

## E    ORDER OF LINEARIZATION

Prior work reports that performance can improve when the generation order of the output sequence is designed during Transformer training (Shen et al., 2023; Sato et al., 2025). Accordingly, we varied the generation order by linearizing QR codes into one-dimensional sequences according to the four orders shown in Table 1, using (v3, L)-QR codes with mask pattern 0. As a result, the ordering in Table 1(d) achieved the highest success rate. This is likely because the ordering in Table 1(d) aligns more closely with the bit ordering of QR codes, as illustrated in Figure 5.

## F    SENSITIVITY OF QR CODES TO PLAIN TEXT CHANGES

In this section, we evaluate the sensitivity of QR code decoding. The sensitivity of a function is defined by how much the output changes in response to variations in the input. In this experiment, we examine how many bits in a QR code must be altered to produce a one-character change in the expected output (i.e., the plain text). We prepare a set of domain names and generate 1,000 samples by randomly modifying a single character in each (e.g., changing "example.com" to "exomple.com"). We then convert both the original and modified strings into QR codes. Finally, we count the number of differing bits between the original and modified QR codes and compute the average.

Table 7 shows the average number of bit changes caused by a one-character change in plain texts for QR codes in different versions. For Version 1 and Version 2, the number of bit changes required to alter the output increases with the error correction level.

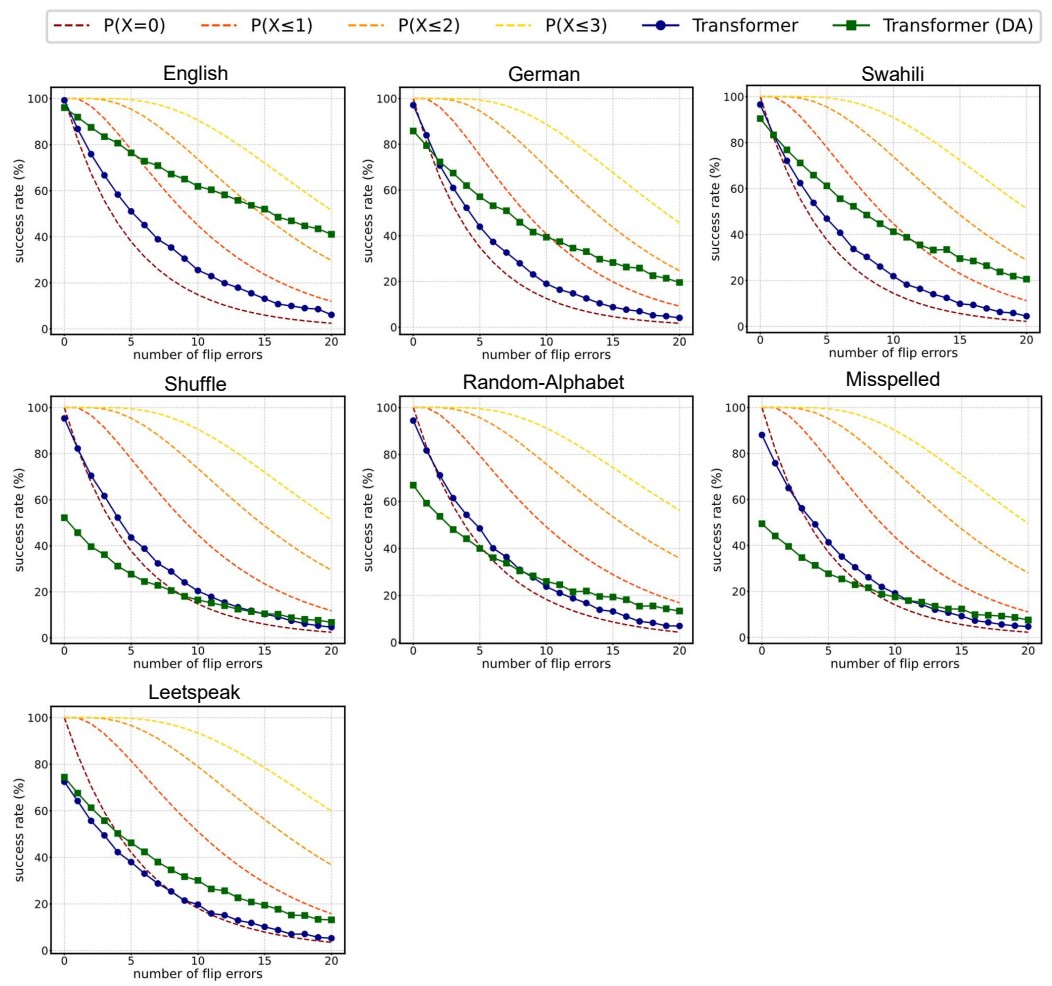

Figure 6: Probabilities (%) that the payload codewords (i.e., the subset of the data codewords that directly encode the plain text) contain at most $X$ errors for $X \in \{0, 1, \dots, 3\}$, alongside the empirical success rates on each dataset defined in Section 6.

This suggests that the decoding of QR codes in Version 1 and Version 2 becomes less sensitive as the error correction level increases. On the other hand, Version 3 does not follow this trend. This is attributed to differences in the encoding schemes used in each version. In Version 3 and later versions, the data codewords are divided into two or more blocks, and each block is independently encoded with Reed–Solomon codes to generate corresponding error correction codewords. Because Version 3 and later versions perform encoding on split data blocks, the function sensitivity does not necessarily decrease with higher error correction levels. Given these factors, Version 1 and Version 2 are more suitable for examining how function sensitivity impacts the behavior of Transformers.

Table 7: Average number of bit changes caused by a single-character change in the plain text.

| Version | L | M | Q | H |
|---|---|---|---|---|
| 1 | 30.3 | 42.0 | 54.1 | 69.2 |
| 2 | 43.4 | 66.7 | 90.2 | 115.4 |
| 3 | 62.1 | 106.5 | 74.3 | 92.1 |

## G  EXAMPLES OF DOMAIN NAME FROM TRANCO

Listing 1 shows the top 30 popular domain names from the Tranco (Pochat et al., 2018) ranking (as of November 27, 2024). It can be seen that many English words are included.

Listing 1: Top 30 domains from the Tranco list

```
google.com
mail.ru
microsoft.com
facebook.com
dzen.ru
apple.com
root-servers.net
amazonaws.com
youtube.com
googleapis.com
akamai.net
twitter.com
instagram.com
cloudflare.com
azure.com
a-msedge.net
gstatic.com
office.com
akamaiedge.net
linkedin.com
live.com
tiktokcdn.com
googletagmanager.com
googlevideo.com
akadns.net
amazon.com
doubleclick.net
windowsupdate.com
fbcdn.net
googleusercontent.com
```

## H    COLLECTING RANDOM WORDS.

For the *English*, *German*, *Swahili* evaluation sets, we used ChatGPT-4o to collect random words and validate them using the Natural Language Toolkit Bird & Loper (2004). Specifically, for each set, roughly 100 words of various lengths were collected. For *English*, the collected words were validated if they were in the NLTK dictionary. For others, the collected words were validated if they were *not* in it. The list of the collected words can be found in the word_list.py in the supplementary material.

## I    EXAMPLES OF UNREADABLE QR CODES (BUT TRANSFORMERS CAN)

Figure 7 present several examples of severely currupted QR codes that are unreadable by pyzbar but readable by Transformers. The setup follows that of Section 5.3. Readers may try to read them using their smartphone.

## J    EXAMPLES OF GENERATED STRINGS.

Examples of strings generated by the Transformer from clean QR codes are shown in Table 8. When the generated string matches the original string precisely, the model successfully handled both short and long plain text. In addition, even when the generated strings did not precisely match the original string, they were generally close to the ground truth. For instance, for "dtv2009.gov," the model outputs "dtv2008.gov," which differs only by one character. There are also cases where the model makes mistakes because it has learned specific patterns in English words. For example, for the plain text "doggettinc.com," the Transformer incorrectly generated "doggetting.com." This was likely because it misinterpreted "inc" as part of the preceding word and segmented it as "getting," reflecting a misunderstanding influenced by learned linguistic regularities. On the other hand, Table 9 shows

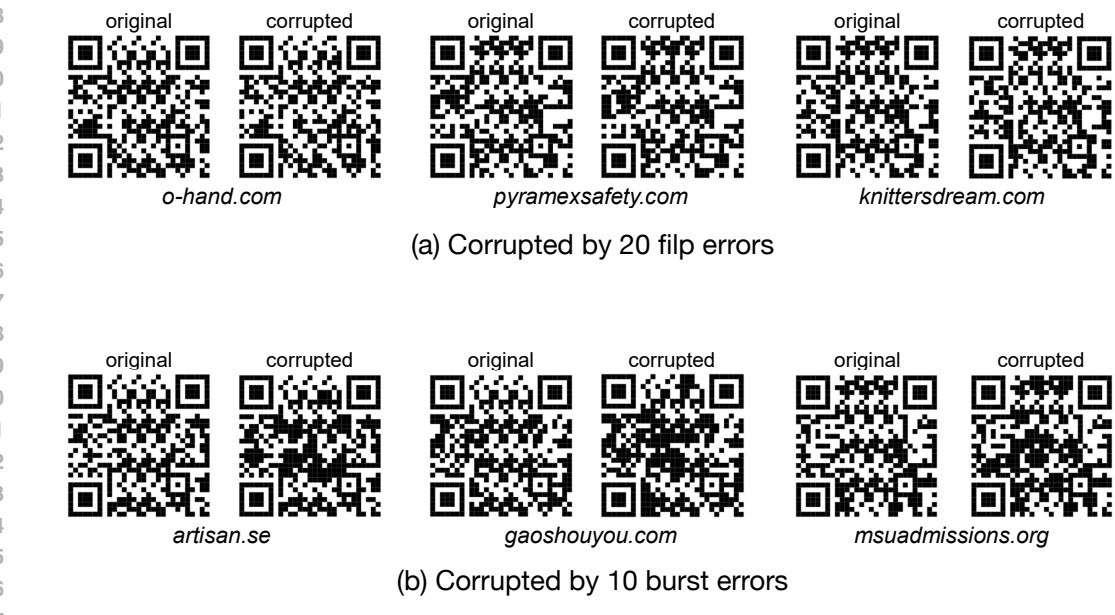

(a) Corrupted by 20 filp errors

(b) Corrupted by 10 burst errors

Figure 7: Examples where the Transformer successfully decodes QR codes (a) with 20 flip errors and (b) 10 3x3 burst errors. "original" shows the QR code before corruption, and "corrupted" shows the version after applying the errors. While `pyzbar` fails to decode the corrupted QR code, the Transformer can decode it.

Table 8: Examples of decoding by Transformers from clean (v3, L)-QR codes (mask pattern 0). The model reconstructs the plain text with a high success rate for both short and long sequences. Even when decoding fails, the generated strings exhibit high similarity to the ground truth.

|  | Ground Truth | Prediction |
|---|---|---|
| ✓ | ellis.ru | ellis.ru |
| ✓ | mobile-arsenal.com.ua | mobile-arsenal.com.ua |
| ✓ | osakabasketball.jp | osakabasketball.jp |
| × | doggettin**c**.com | doggettin**g**.com |
| × | elalman**a**que.com | elalman**i**que.com |
| × | dtv200**9**.gov | dtv200**8**.gov |

examples of outputs generated from inputs with 20-bit flips. Compared to the clean examples in Table 8, the outputs are generally less similar to the original strings. However, some errors affect only one character, such as "delicom.global" becoming "dedicom.global." As shown in Figure 2(a), `pyzbar` fails to return any output when 20-bit corruption is applied. This is because it is designed to suppress output entirely once the corruption exceeds a certain threshold to avoid incorrect results. In contrast, the Transformer always produces an output, even if it is incorrect. While this can lead to errors, the outputs often remain close to the original, even with heavy corruption. Additionally, in Figure 3, we present the distribution of similarity between the generated strings and the original plain text under severe corruption.

## K    DISCUSSION ON THE TRANSFORMER'S ROBUSTNESS TO CORRUPTION

As shown in Figure 2, the Transformer continues to decode accurately even under heavy corruption, sometimes exceeding the theoretical error-correction limit. Two key factors appear to underlie this robustness: exploiting word structure and a focus on data codewords.

Table 9: Examples of decoding by Transformers from corrupted (v3, L)-QR codes (mask pattern 0). For inputs with 20-bit flips, lower-similarity failures are observed. However, despite severe corruption, both failure cases are also similarity to the ground truth.

|   | Ground Truth | Prediction |
|---|---|---|
| ✓ | casino-x-dawn.bet | casino-x-dawn.bet |
| ✓ | mobil-isc.de | mobil-isc.de |
| ✓ | slotsyps.info | slotsyps.info |
| ✗ | de**l**icom.global | de**d**icom.global |
| ✗ | inh**and**.com | inh**cdn**.com |
| ✗ | mob**i**le-arsena**l**.com.**ua** | mob**x**le-arsena**t**.com.**mt** |

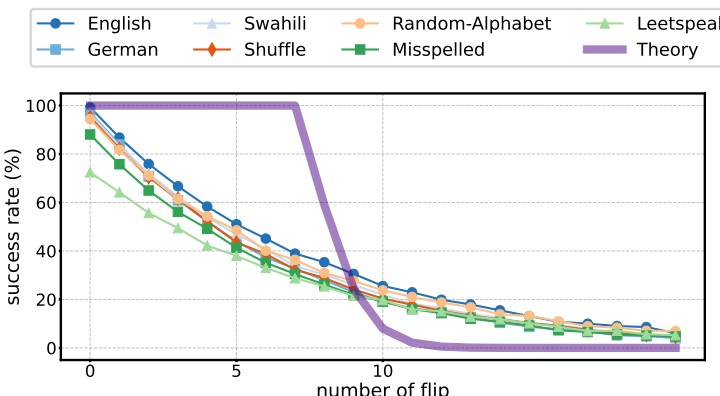

Figure 8: Decoding success rates (%) under flip errors on each dataset defined in Section 6. As corruption increases, the ranking broadly mirrors Table 5.

The main factor is that the Transformer exploits word structure during inference, even under corruption. Figure 8 shows decoding success rates (%) under flip errors on each dataset defined in Section 6. As corruption increases, the ranking broadly mirrors Table 5: the Transformer performs best on datasets with word structures (i.e., English, German, Swahili), worse on datasets without such structures (i.e., Shuffle, Random-Alphabet), and worst on datasets that preserve word structures but include corruptions (i.e., Misspelled, Leetspeak). These results suggest that the Transformer leverages word structure during inference, which in turn supports more robust decoding under corruption.

In addition, as illustrated in Figure 4(b), the Transformer's tendency to focus primarily on the data codewords can also be considered a second factor. Standard QR readers, constrained by Reed–Solomon coding, fail once the number of errors in the error-correction codewords exceeds a fixed threshold. By contrast, because the Transformer attends mainly to data codewords, corruption in the error-correction codewords does not directly impair its predictions. Consequently, when error-correction codewords are corrupted, the Transformer has an advantage over standard QR readers. Importantly, this does not mean the Transformer performs zero error correction. Figure 9 compares the empirical success rates with the probabilities that the subset of data codewords that directly encode the plain text (hereafter, *payload codewords*) contains no errors, at most one error, at most two errors or at most three errors. If the Transformer corrected nothing and succeeded only when the payload codewords were entirely error-free, its success rate would not exceed the probability of having no errors. In practice, the Transformer often succeeds even when the payload codewords contain a small number of errors. Its success rate is higher than the probability of having no errors at all. This suggests that the Transformer performs some level of error correction. As illustrated in Figure 6, we observe the same pattern for every dataset in Section 6. However, Figure 6 also shows that the gap between the Transformer's success rate and the probability of having no errors in the payload codewords differs across datasets. This result indicates that the Transformer's use of word structure, identified as the primary factor, also plays a role. These observations do not

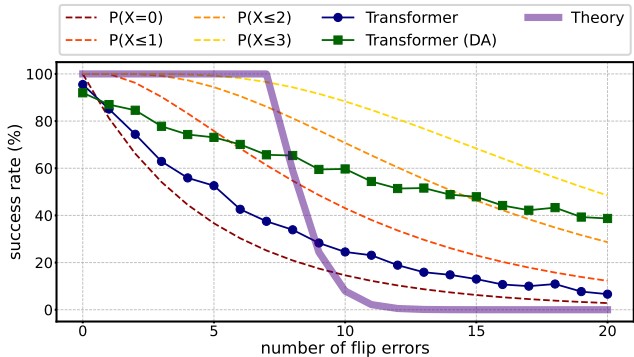

Figure 9: Probabilities (%) that the payload codewords contain no errors, at most one error, at most two errors or at most three errors, alongside the empirical success rates. Payload codewords are a subset of the data codewords that directly encode the plain text. $X$ denotes the number of flip errors occurring in the payload codewords. The Transformer's success rate is generally higher than the probability that the payload codewords contain no errors.

constitute a complete account of robustness. Future work will clarify the remaining factors and their interactions.

