# OpenReview forum: "Learning Medium-Sensitivity Functions: A Case Study on QR Code Decoding"
_ICLR.cc/2026/Conference — Submitted to ICLR 2026_

### Official Review · Reviewer_pS7C · 2025-10-30

**Soundness:** 3
**Presentation:** 4
**Contribution:** 3
**Rating:** 4
**Confidence:** 4

**Summary:**

The paper studies how Transformers perform on QR code decoding. The authors first train a model to identify the mask pattern and then use a Transformer to decode the QR data. They find that Transformers remain robust under bit-flip noise even without using error-correction codes, achieving performance beyond the theoretical limit. The authors suggest that this robustness comes from the model’s ability to capture language structures, as shown by the variation in decoding success across different languages. This implies that Transformers may correct some intentional misspellings in QR codes.

**Strengths:**

The paper explores an interesting and relatively unexplored task. The authors evaluate the model’s performance across different mask patterns and provide an insightful analysis of why Transformers can surpass classical decoding algorithms and even exceed theoretical performance limits.

**Weaknesses:**

The task is interesting, but I am concerned about the level of difficulty. Since Reed–Solomon doesn’t modify the original URL codewords, a Transformer is effectively learning to ignore irrelevant components (function patterns, err correction words) and to map 8-bit codewords back to characters. Its apparent error-correction ability comes from modeling language structure rather than true parity-based correction.

**Questions:**

(1) What's the architecture/size of the mask-classification model?
(2) Is a 12-layer Transformer unnecessarily large for retrieving a short URL, which typically consists of only a few words or tokens? A smaller or shallower model might achieve similar performance with far less computational cost.

---

> ### Author Response · Authors · 2025-11-24
> **Official Comment by Authors (1/2)**
>
> We sincerely thank the reviewer for the thoughtful and constructive feedback. Below, **Key Responses** presents an overview of the focus of discussion and quick answers to them. Then, the detailed answers follow.
>
> ---
>
> ### **Key Responses**
>
> The reviewer might not fully grasp the focus of this work. The key focus and contributions
> - **The QR code decoding task.** The QR code task is not a hard task itself, but it was adopted as a showcase of learning medium-sensitivity functions. Thus, the concern of the level of difficulty is slightly off the point. Note that the task is not necessarily learning Reed--Solomon decoding, either.
> - **The core contributions.** The critical observations are that i) Transformer learns QR code decoding in its own way of using language structures from English-based URLs (thereby attaining robust decoding), and ii) surprisingly, this generalizes to other languages and even random alphabetic strings (i.e., general QR code reading is achieved in a sense).
> - **Practicality.** The focus of this work is not on developing AI-based QR code readers, although our results strongly suggest this as a promising direction. Thus, potential reduction of computational cost using smaller/shallower models is not critical.
>
> Overall, we currently do not identify any critical weaknesses from the reviewer comments. We hope this rebuttal clarifies the focus of the discussion, thereby enhancing the reviewer's understanding and reconsideration of this work.

---

> ### Author Response · Authors · 2025-11-24
> **Official Comment by Authors (2/2)**
>
> ### **Level of difficulty**
> > I am concerned about the level of difficulty [...], a Transformer is effectively learning to ignore irrelevant components [...] and to map 8-bit codewords back to characters.
>
> The QR code decoding task itself is not a hard task, and addressing hard tasks is not our focus. It is common to examine Transformers' learning behavior with a small, restricted task. For example,
> - [1] discovered the well-known *grokking* through modular arithmetics, followed by many studies on this phenomenon.
> - [2] demonstrated the impact of decoder-input order through two-integer addition. Our Table 1 is also inspired by their work, and recently, there has been an attempt to discover task-specific learning-friendly order automatically [3].
> - [4] addressed the integer addition task, discovering additional position encoding to emphasize the correspondence between digits, enhancing the length generalization.
> - [5] reported the effect of base using Integer GCD. Namely, tokenizing each integer in a particular base (e.g., 7 to 111 for base 2) has a striking impact on the learning.
>
> There are all seminal works, focusing on one for a few tasks, with inspiring observations. To the best of our knowledge, there is no such well-focused case study (at least explicitly) when it comes to learning medium-sensitivity. We adopted QR code decoding and discovered the unexpectedly robust decoding ability.
>
> A critical difference from prior studies is that our work targets medium-sensitivity functions, while the aforementioned works handled input-sensitive functions (although no clear discussion was given from this perspective). We consider that QR code decoding is a good non-artificial example with a clear application, giving an intuition of the needs of sensitivity and insensitivity at decoding.
>
> [1] Power, "Grokking: Generalization Beyond Overfitting on Small Algorithmic Datasets," arXiv:2201.02177, 2022.
>
> [2] Shen et al., "Positional description matters for transformers arithmetic" arXiv:2311.14737, 2023.
>
> [3] Sato et al., "Chain of Thought in Order: Discovering Learning-Friendly Orders for Arithmetic," ICML MATH4AI Workshop, 2025
>
> [4] McLeish et al., "Transformers Can Do Arithmetic with the Right Embeddings", NeurIPS, 2024
>
> [5] Charton, "Learning the greatest common divisor: explaining transformer predictions," ICLR, 2024.
>
> ---
>
> ### **Source of robustness**
> > Its apparent error-correction ability comes from modeling language structure rather than true parity-based correction.
>
> Yes, this is the core observation of our work. The derived theoretical bound and Figure 2 show that using language structure, the decoding can surpass the theoretical limit. Most of the other experiments are dedicated to explaining this from various aspects. Notably,
> - Transformers acquire robust decoding not through error-correction bits (Figure 4).
> - Still, Transformers are not memorizing language (particularly, English) structure; it generalizes to other languages and even to random alphabetic strings (Table 5).
>
> Note that our task is QR code decoding and not Reed--Solomon decoding. Initially, we expected that the Transformer would learn the Reed--Solomon decoding (i.e., upper bounded by the theoretical limit), but the experiments revealed an interesting consequence.
>
> Recently, there has been a new trend of **learned** data structures [6], where significantly efficient random access (e.g., `a[i]` or `13 in a?` for an enormous array `a`) is attained by exploiting the distribution of stored data. Our QR code decoding could also be discussed from this perspective. Namely, one may assume some structure in the embedding texts of QR codes and exploiting it for robust decoding.
>
> [6] Al-Mamun et al., "A Survey of Learned Indexes for the Multi-dimensional Space," ACM Computing Surveys, 2025
>
> ---
>
> ### **Questions**
>
> > (1) What's the architecture/size of the mask-classification model?
>
> All the experiments are conducted with a Transformer model with six encoder and six decoder layers (see Sec. 5.1). The mask classification was performed with the encoder-only version of this model with a linear classification head. We will clarify this in Sec. D at the revision.
>
>
> > (2) Is a 12-layer Transformer unnecessarily large for retrieving a short URL, which typically consists of only a few words or tokens? A smaller or shallower model might achieve similar performance with far less computational cost.
>
> We used the 6/6-Transformer as it is one of the standard architectures and has sufficient learning capacity. Smaller/shallower models can attain similar results with less computational cost. As we have already mentioned, this computational efficiency is not the focus, and thus we were based on a standard model with a sufficient capacity. Instead, we tested many combinations in QR codes (i.e., 8 mask patterns, 3 versions, 4 levels), input orders (4 patterns; Table 1), data augmentation, and OOD evaluation sets (6 sets; Table 5).

---

> > ### Comment · Reviewer_pS7C · 2025-11-25
> >
> > **The weakness is that the work relies on a single task that is neither sufficiently general nor challenging, and it does not provide a solid contribution to deep learning.**
> >
> > The weakness is not just about QR decoding is not a hard task, but rather about whether the single chosen task is sufficient to justify the broader claims about “medium-sensitivity functions.” If the paper’s goal is to define and study medium-sensitivity functions, limiting the entire evaluation to QR decoding (plus synthetic variants) does not convincingly support the claimed scope.
> >
> > I do not think the listed grokking literature and the arithmetic papers should be viewed as mere case studies. They are important works that uncover genuinely interesting training dynamics, or they design and solve existing real problems. The *grokking modular arithmetic* paper identified the minimal model depth at which grokking emerges and proposed complete interpretability of the solution employed by the model; *Transformers Can Do Arithmetic with the Right Embeddings* tackles large-number addition and introduces the abacus embeddings; these contributions go far beyond a simple case analysis.
> >
> > From the perspective of learned data structures, given that 12/(6+6)-layer Transformers are used throughout this task, I would expect at least a study of minimal model depth and width. If the paper aims to analyze learned representations, it should then interpret the representations of the smallest model that successfully solves the task.
> >
> > After reading the other reviews and the citations they provided, I do not believe this submission has the level of novelty claimed in the paper. It needs additional contributions, either by incorporating more medium-sensitivity tasks or by providing a solid contribution in interpretability or other relevant area in deep learning. I think the current submission does not reach the level of contribution expected for a poster at a major deep learning conference.

---

### Official Review · Reviewer_qpnk · 2025-10-31

**Soundness:** 3
**Presentation:** 3
**Contribution:** 2
**Rating:** 4
**Confidence:** 3

**Summary:**

This paper presents a case study on learning medium-sensitivity functions through the task of QR code decoding. The authors construct several URL datasets and employ a standard Transformer architecture and training strategy. Experimental results show that Transformer-based QR code decoding achieves a high success rate, demonstrates robustness to various levels of corruption, and generalizes beyond English-rich data to other languages. This study provides insights into the potential of learning-based approaches for medium-sensitivity tasks.

**Strengths:**

1. The paper is well written and easy to follow. The motivation for studying medium-sensitivity functions is clearly articulated.

2. The ablation study is comprehensive, covering different mask patterns and corruption types, including flip and burst errors.

**Weaknesses:**

1. While QR code decoding is an interesting example and the ablation studies are detailed, the overall scope of the paper is narrow. It is unclear how the findings can be generalized to a broader set of medium-sensitivity tasks.

2. The evaluation is limited. The impact of model architecture is not explored, and the test set consists of only 1,000 samples, compared to 500,000 training samples. Increasing the diversity and size of the test set would help validate the generality of the results.

**Questions:**

1. Is it possible to use a hybrid input order? For example, Table 1 shows the effects of row and column ordering. Could a combined strategy, such as block-wise or diagonal ordering, be beneficial? Could the performance differences be attributed to positional embeddings?

2. It is notable that a learning-based method can achieve high success rates in QR code decoding. Are there other real-world applications of medium-sensitivity functions where such methods could be applied? Can the learned model generalize to these other tasks?

---

> ### Author Response · Authors · 2025-11-24
> **Official Comment by Authors (1/3)**
>
> We thank the reviewer for the balanced and constructive comments.
> Below, we address each point in detail and clarify the positioning of our work.
>
> ---
>
> ### **Key Responses**
>
> The reviewer highly evaluated our motivation for studying medium-sensitivity function learning and the comprehensive case study on QR codes. Our quick answers to weaknesses and questions are as follows.
>
> - **Case studies on small, restricted tasks have revealed many insightful observations in the literature of learning high-sensitivity functions (see below)**. Our work aligns with them. Restricting ourselves to QR code decoding (this choice is explained in the Introduction) leads to a comprehensive analysis, discovering unexpected robustness and generalization. As the reviewer highlights, our comprehensive case provides an explanation for these observations.
>
> - The evaluation should be sufficient (but we are open to scaling it upon request). Based on a standard 6/6-Transformer, **our experiments cover many combinations in QR codes (i.e., 8 mask patterns, 3 versions, 4 levels), input orders (4 patterns; Table 1), data-augmentation, and OOD evaluation sets (6 sets; Table 5). This yields 96 trained models (additionally 4 input orders for some of them)**. The OOD evaluation sets have three reasonable categories. Each test set contains 1,000 samples, but scaling them up to 1M must not make an essential change.

---

> ### Author Response · Authors · 2025-11-24
> **Official Comment by Authors (2/3)**
>
> ### **W1. "The overall scope appears narrow."**
>
> > *"While QR code decoding is an interesting example, the overall scope of the paper is narrow. It is unclear how the findings generalize to other medium-sensitivity tasks."*
>
> Our work provides a thorough case study on QR code decoding with several remarks that can impact the broader context from three aspects: (a) positioning in the literature, (b) novel observations in robustness and generalization, and (c) showing the promising potential of learning-boosted QR code readers.
>
> (a), (b) suggests that QR code decoding is one of the potential standard tasks in the underexplored field of learning middle-sensitivity functions, as the parity function is for learning high-sensitivity functions. Learning low-sensitivity functions is easy and well-studied, so the applications are important. The high-sensitivity one is hard, so the artificial base task is convenient. Our medium-sensitivity case lies between them.
> (c) indicates the practical utility. As we discuss below, this can also be viewed from the recently learned data structure.
>
> We elaborate on (a)--(c) below.
>
> ---
> **(a) First case study of learning medium-sensitivity function.**
>
> Note that case studies on small, restricted tasks have been evaluated as independent works with great impact in the literature of learning high-sensitivity functions. For example,
> - [1] discovered the well-known *grokking* through modular arithmetics, followed by many studies on this phenomenon.
> - [2] demonstrated the impact of decoder-input order through two-integer addition. Our Table 1 is also inspired by their work, and recently, there has been an attempt to discover task-specific learning-friendly order automatically [3].
> - [4] proposed an addressed integer addition task, discovering additional position encoding to emphasize the correspondence between digits, enhancing the length generalization.
> - [5] reported the effect of base using Integer GCD. Namely, tokenizing each integer in a particular base (e.g., 7 to 111 for base 2) has a striking impact on the learning.
>
> These are seminal works on small, focused tasks with inspiring observations. To the best of our knowledge, there is no such well-focused case study (at least explicitly) when it comes to learning medium-sensitivity. Thus, we consider this work to reserve an important position in the context of learning functions with various sensitivities.
>
> [1] Power, "Grokking: Generalization Beyond Overfitting on Small Algorithmic Datasets," arXiv:2201.02177, 2022.
>
> [2] Shen et al., "Positional description matters for transformers arithmetic" arXiv:2311.14737, 2023.
>
> [3] Sato et al., "Chain of Thought in Order: Discovering Learning-Friendly Orders for Arithmetic," ICML MATH4AI Workshop, 2025
>
> [4] McLeish et al., "Transformers Can Do Arithmetic with the Right Embeddings", NeurIPS, 2024
>
> [5] Charton, "Learning the greatest common divisor: explaining transformer predictions," ICLR, 2024.
>
> ---
>
> **(b) Discovering non-trivial robustness and generalization.**
>
> Our key experimental results are as follows.
> - Transformer decoding robustness surpasses the theoretical limit of successful decoding under bit flip noises.
> - The sources of robustness are the intrinsic structure of natural languages. Noteably, Transformers learn it from an English-rich dataset, but the decoding is also successful in non-English and even random alphabetical strings, indicating strong generation with robustness.
>
> I assume that the reviewer follows them, but we are open to elaborating more as necessary.
>
> ---
>
> **(c) Suggesting learning-boosted QR code readers.**
>
> Learning-based QR code itself is novel in the literature. This may attract independent interests from practitioners. Particularly, the experiments show that the robustness of Transformer decoding is based on the intrinsic structure of natural languages and thus orthogonal to the classical error-correction mechanism. This strongly suggests the hybrid methods as a future work.
>
> Recently, there has been a new trend of **learned** data structures [6], where significantly efficient random access (e.g., `a[i]` or `13 in a?` for an enormous array `a`) is attained by exploiting the distribution of stored data. Our QR code decoding could also be discussed from this perspective. Namely, one may assume some structure in the embedding texts of QR codes and exploiting it for robust decoding. We will revisit this topic in the answer to Q2.
>
> [6] Al-Mamun et al., "A Survey of Learned Indexes for the Multi-dimensional Space," ACM Computing Surveys, 2025

---

> ### Author Response · Authors · 2025-11-24
> **Official Comment by Authors (3/3)**
>
> ### **W2. "Limited evaluation."**
>
> > *"The evaluation is limited. The impact of model architecture is not explored, and the test set consists of only 1,000 samples, [...]. Increasing the diversity and size of the test set would help validate the generality of the results."*
>
> We appreciate it if the reviewer could provide more elaboration on "the diversity and size of the test set."
>
> As for sample size, our experiments cover many combinations in QR codes (i.e., 8 mask patterns, 3 versions, 4 levels), input orders (4 patterns; Table 1), and **each** has 1,000 samples (see Table 3). It is hard to believe that increasing 1,000 to 1M brings meaningful change.
>
> As for diversity, our experiments include 1 in-distribution and 6 out-of-distribution evaluation sets (Table 5). Transformers are trained on English URLs, and then tested on the evaluation sets with significantly different structural/statistical patterns in numerics and alphabets.
>
> We welcome the opportunity to conduct more experiments if specified with reasonable motivations. It should improve our work.
>
>
> ---
>
> ### **Q1. "Is hybrid input ordering (block-wise, diagonal, etc.) beneficial?"**
>
> > Is it possible to use a hybrid input order? For example, Table 1 shows the effects of row and column ordering. Could a combined strategy, such as block-wise or diagonal ordering, be beneficial? Could the performance differences be attributed to positional embeddings?
>
> Thank you for the insightful questions.
>
> The hybrid combinations can improve the accuracy and worth of testing. Tables 1 and 5 show that the accuracy is almost saturating, so the impact is expected to be limited unless the QR code size scales up significantly (but this leads to large memory consumption due to attention).
>
> Positional embeddings must have a limited impact. Note that the ordering here is about the target sequence (i.e., decoder input). Several studies show that the removal of position encoding from decoder input has a limited impact on learning because the auto-regressive generation naturally induces an ordering [7, 8]. Thus, the ordering of auto-regressive generation (i.e., target sequence) matters more. Its impact is also shown in [2] for the integer multiplication task and in [3] for more general cases.
>
> [7] Kazemnejad et al., "The Impact of Positional Encoding on Length Generalization in Transformers," NeurIPS'23.
>
> [8] Zuo et al., "Position Information Emerges in Causal Transformers Without Positional Encodings via Similarity of Nearby Embeddings," COLING, 2025
>
> ---
>
> ### **Q2. "Other real-world applications of medium-sensitivity functions"**
> > It is notable that a learning-based method can achieve high success rates in QR code decoding. Are there other real-world applications of medium-sensitivity functions where such methods could be applied? Can the learned model generalize to these other tasks?
>
> Yes. Actually, well-known real-world applications range from low to high-sensitivity cases, so the utility of learning-based methods is already proven. The case study in high-sensitivity literature and our medium-sensitivity case study try to understand them at a high level.
>
> I suppose that the reviewer's interest aligns with recent **learned** data structures [1], where a learning-based method is introduced, e.g., for efficient indexing (random access) of a huge array. Their critical assumption for acceleration is that the real-world data to store is not random but has some particular distributions that can be modeled by deep learning models. Similarly, in our case, we focus on a typical QR code usage, URL embedding. Note that the learned data structure is more focused on efficiency, but our results suggest robustness.
>
> We were originally not aware of this interesting similarity and contrast with learned data structures. We appreciate the reviewer's inspiring question and will include this in the revision.

---

### Official Review · Reviewer_v7oJ · 2025-11-02

**Soundness:** 2
**Presentation:** 2
**Contribution:** 1
**Rating:** 2
**Confidence:** 3

**Summary:**

This submission is about using transformers to decode QR codes. The overall takeaway is that the models learn to do this extremely well.

The work is pitched as the first study into learning what the authors term *medium-sensitivity* functions. This does not seem to be formally defined, but they compare with parity learning (high sensitivity) and image classification (which they call low sensitivity).

**Strengths:**

The experiments seem interesting and I expect this work will be turned into a solid contribution in the future.

**Weaknesses:**

The authors have missed key prior work. I found a blog post [1] and Github repo [2] on learning neural networks to decode QR codes specifically.

I am not an expert, but there appears to be decades of work on using neural networks to decode error-correcting codes [3,4] and more recent work that includes training [5,6].

It seems that a major revision is required.

[1] [https://medium.com/@andrewromanenco/decoding-qr-codes-using-neural-networks-272f9e8ba635](https://medium.com/@andrewromanenco/decoding-qr-codes-using-neural-networks-272f9e8ba635)

[2] [https://github.com/Brainydaps/AI-QR-Code-Decoder](https://github.com/Brainydaps/AI-QR-Code-Decoder)

[3] Yuan, Jing, and C. S. Chen. "Neural net decoders for some block codes." IEE Proceedings I (Communications, Speech and Vision) 137.5 (1990): 309-314.

[4] Yuan, Jing, V. K. Bhargava, and Q. Wang. "An error correcting neural network." Conference Proceeding IEEE Pacific Rim Conference on Communications, Computers and Signal Processing. IEEE, 1989.

[5] Beery, Yair, David Burshtein, and Eliya Nachmani. "Deep learning decoding of error correcting codes." U.S. Patent Application No. 15/996,542.

[6] Choukroun, Yoni, and Lior Wolf. "Error correction code transformer." Advances in Neural Information Processing Systems 35 (2022): 38695-38705.

**Questions:**

Can you give a mathematical definition of "medium-sensitivity function"?

---

> ### Author Response · Authors · 2025-11-24
> **Official Comment by Authors**
>
> We sincerely thank the reviewer for the time and effort spent evaluating our paper and for raising concerns regarding novelty and related work. We appreciate the opportunity to clarify our contributions and to situate our work more clearly within the existing literature.
>
> ---
>
> ### **Key responses**
>
> The reviewer poses only one weakness---not citing several "key prior works." Six sources [1]–[6] are given. We respectfully argue that none of them diminishes the novelty of our work, for three concise reasons:
>
> 1. **Non-scientific sources ([1], [2], [5]) are not valid prior research.**
>  They are not peer-reviewed, no theoretical analysis, no systematic experiments, and no connection to our contributions. It is also confusing what makes the reviewer put blog post [1] and GitHub repo [2] in the top of the list of "key prior works."
>
> 2. **Classical NN-based ECC works [3], [4] from 1989/1990 target small block codes using small neural nets.**
>  These are fundamentally different from our formulation, scale, or goals. Even our theoretical results are not covered by them.
>
> 3. **The NeurIPS 2022 paper [6] addresses ECC decoding of algebraic codes.**
>  Our work studies *medium-sensitivity functions*, where robustness arises from **semantic redundancy**, not parity.
>  The problem setting, inputs/outputs, scientific questions, and contributions do not overlap (see below).
>
> For these reasons, we strongly believe that our novelty remains intact, although we will cite [6] (and perhaps [3,4]) in the revision for completeness.
> We kindly ask the reviewer to elaborate on the novelty concern if any element of our theoretical or empirical contributions appears missing. Below, we elaborate on the second and third points.
>
> ---
>
> ### **Comparison to [3,4].**
>
> These works use small feed-forward networks applied to simple linear block codes (Hamming/BCH), with datasets of only a few thousand samples. Our work employs high-capacity Transformers trained on millions of QR codes, involving mask patterns, format information, Reed–Solomon structure, and natural-language outputs. Their setting does **not** cover any aspect of our contributions (theory, sensitivity analysis, semantic generalization).
>
> ---
>
> ### **Comparison to [6]**
>
> Figure 1 in [6] immediately tells that [6] and our work address distinct work.
> The former models noisy transmission, where the bits are perturbed by Gaussian (i.e., real-value) noise, and the decoder aims at retrieving the original signals.
> As the Introduction in [6] writes, this problem has been known as an NP-hard problem. The modeling and motivation in [6] are strongly biased to real-world bit string transmission through noisy channels. Their work addresses linear codes; the main focus is on the network architecture, and any of our empirical results are relevant.
>
> ---
>
> ### Q1. Mathematical definition of "medium-sensitivity function"
>
> Medium-sensitivity functions refer to those that need at least a moderate number of input changes to change their output. The sensitivity ranges from low to high continuously so there are not clear-cut threshold of "medium" level. The sensitivity can be formally defined as follows.
>
> **Definition (Local sensitivity).**
> For a function $f : \mathcal{X} \to \mathcal{Y}$ on a discrete domain with distance $d_{\mathcal{X}}$,
> its *local sensitivity* at $x$ is
> $${\rm LS}(f,x) = \max_{x' : d_{\mathcal{X}}(x,x') = 1} d_{\mathcal{Y}}(f(x), f(x')).$$
>
> **Definition (Global sensitivity).**
> The *global sensitivity* of $f$ is
> $${\rm GS}(f) = \mathbb{E}_{x \in \mathcal{X}} {\rm LS}(f,x).$$
>
> Our study does not mathematically handle the sensitivity, but we agree that including mathematical definition will make the target of discussion clear. We thank the reviewer and will update the manusript accordingly.

---

> > ### Comment · Reviewer_v7oJ · 2025-11-24
> >
> > Thank you for your response. After reading the other reviews, my impression of the work remains unchanged. I will reiterate my two main points.
> >
> > First, it is important to cite all prior work on a topic, even if that work has not appeared in a scientific venue. (By all means, point out to the reader that it is not peer-reviewed!) Work on using neural networks for error-correcting codes is absolutely relevant, even if the focus is different. I spent only a few minutes finding those references; you need to spend more time than that.
> >
> > Second, to evaluate the claim "this study provides the first case study of learning medium-sensitivity functions," we need to know what a medium-sensitivity function is. (I suspect that, if you did write down a definition, we would be able to find existing literature on learning functions that satisfy it.)

---

> ### Author Response · Authors · 2025-11-24
> **Official Comment by Authors**
>
> We thank the reviewer for the follow-up and for clarifying the two outstanding concerns.
> Below we respond carefully and concisely.
>
> ---
> ### **(1) On references**
>
> > "It is important to cite all prior work, even if not peer-reviewed."
>
> We understand the reviewer's point. In the revision, we will cite the suggested items [1–6] for completeness, while clearly noting their scientific status (e.g., blog/GitHub vs. peer-reviewed papers).  Our intention was not to omit them, but to avoid overclaiming connections where the problem settings differ substantially. We appreciate the clarification.
>
> Importantly, the reviewer did *not* indicate how these works invalidate our contributions, and we believe our paper's claims remain intact once the citations are added.
>
> ---
> ### **(2) On defining medium-sensitivity functions**
>
> > "To evaluate the claim, we need to know what a medium-sensitivity function is."
>
> We agree that making the definition explicit will help readers. In the rebuttal we provided the definition of sensitivity (particularly, global sensitivity).  The "medium" cannot be clearly defined as low/high-sensitiviy cannot. Our use of “medium-sensitivity” simply denotes functions whose sensitivity lies between the two extremes:
>
> - **low-sensitivity:** small input changes rarely change output (e.g., image classification)
> - **medium-sensitivity:** moderate—but not minimal—input changes are required to alter the output (e.g., QR codes)
> - **high-sensitivity:** a single or a few bit changes can alter the output (e.g., parity)
>
> Conceptually, we have two thresholds $\theta_1$ and $\theta_2$. Suppose a function has sensitivity $\theta$ (e.g., according to the definition given in our rebuttal). Then, this function is low-sensitivity if $\theta < \theta_1$, medium-sensitivity if $\theta_1 \le \theta < \theta_2 $, and high-sensitivity if $\theta_2 \le  \theta$. However, it is hard to pinpoint $\theta_1, \theta_2$.
>
> For example, there are many papers that  "low-frequency" components of signals/images. However, they don't define *low* frequency (but, frequency has a clear definition). A closer example is [1].
>
> [1] Hahn et al., "Why are Sensitive Functions Hard for Transformers?" ACL, 2024
>
> This work discusses the difficulty of learning high-sensitivity functions. They provides the definition of sensitivity, but no clear definition (i.e. threshold) of *high* sensitivity is presented.
>
> We will revise the paper to make this explanation clearer.
>
> ---
>
> ### **Summary**
>
> We appreciate the reviewer's clarification. We will include the suggested references and incorporate a clearer definition and discussion of sensitivity. While our scientific conclusions remain unaffected, we thank the reviewer for helping us improve the presentation.

---

### Official Review · Reviewer_eM4m · 2025-11-05

**Soundness:** 3
**Presentation:** 4
**Contribution:** 3
**Rating:** 6
**Confidence:** 3

**Summary:**

The paper presents a rigorous empirical and theoretical study of learning medium-sensitivity functions through Transformer-based QR code decoding. It formalizes QR decoding as an intermediate sensitivity task—sensitive to text changes but robust to bit-level noise—and derives a novel analytical model for the Reed–Solomon success rate under corruption. Experiments demonstrate that Transformers not only exceed the theoretical error-correction limits but also generalize across languages and random strings, exploiting statistical regularities of natural words.
This work bridges sensitivity theory and deep learning, revealing how attention models perform hybrid symbolic–robust computations.

**Strengths:**

- Introduces a novel conceptual framework of medium-sensitivity learning, formalizing an intermediate regime between robustness and semantic invariance in neural models.

- Provides a theoretically grounded link between Transformer behavior and Reed–Solomon coding limits, extending information-theoretic analysis to neural error correction.

- Presents quantitative evidence that Transformers can exceed classical decoding thresholds, revealing emergent redundancy exploitation beyond explicit code design.

- Demonstrates broad empirical validity across synthetic and linguistic datasets, supporting both the theoretical model and its generalization to real-world settings.

**Weaknesses:**

- The theory assumes idealized noise and independence, which may not reflect real Transformer behavior.

- Experiments are limited in scale and domain, so generalization beyond controlled settings is unclear.

- Computational cost and scalability are not discussed, leaving practical applicability uncertain.

**Questions:**

1. How does your medium-sensitivity framework connect to known notions of robustness or Lipschitz continuity, and can it be quantified by model capacity or gradients?

2. How does the mathematical formulation of medium-sensitivity generalize beyond the discrete symbol space—can the same definitions and bounds be extended to continuous vector spaces where distance and corruption are not count-based?

3. Can your Reed–Solomon analysis handle non-linear or data-dependent noise, and how would that change the theoretical limits?

4. What kinds of failure cases did you observe, and what do they reveal about the limits of medium-sensitivity learning?

---

> ### Author Response · Authors · 2025-11-24
> **Official Comment by Authors (1/2)**
>
> We sincerely thank the reviewer for the exceptionally thoughtful and constructive feedback.
> We were very encouraged that the reviewer clearly understood the conceptual motivation of our work, the role of medium-sensitivity, and the theoretical–experimental bridge we aimed to build. Below, we elaborate on the reviewer’s questions and suggestions, which will help us further strengthen the final version.
>
> ---
> ### **W1. Noise design**
> > The theory assumes idealized noise and independence, which may not reflect real Transformer behavior.
>
> There could be some misunderstanding. The noise design on QR codes does not model Transformer behavior (e.g., internal processing error). We introduce the random flip noise as it naturally aligns with the concept of function sensitivity. The burst noise, on the other hand, is not independent noise; once a single bit flip occurs, the bits around it also flip. This mildly models the real-world scenario. A more realistic scenario needs to take into account QR code **readers/detectors** (not our decoder part). Namely, QR code readers read out a QR code from a possibly small, blurry image. Prior works address this step (see Related Work section), but reading errors may remain. For practical use, we need to model the distribution of such errors, but this is beyond the scope of this study.
>
> The next weaknesses also relate
>
> ---
> ### **W2. Experiment scale / W3. computational cost**
> > Experiments are limited in scale and domain, so generalization beyond controlled settings is unclear.
> > Computational cost and scalability are not discussed, leaving practical applicability uncertain.
>
> We acknowledge these weaknesses (with some remarks), but in short, practicality is not the focus but rather a bonus of this work. Our work provides in-depth a case study of learning medium-sensitivity functions. Comparisons with the literature of learning medium-sensitivity may assist reviewer comprehension of our work and the context.
>
> Case studies on small, restricted tasks have been evaluated as independent works with great impact in the literature of learning high-sensitivity functions. For example,
> - [1] discovered the well-known *grokking* through modular arithmetics, followed by many studies on this phenomenon.
> - [2] demonstrated the impact of decoder-input order through two-integer addition. Our Table 1 is also inspired by their work, and recently, there has been an attempt to discover task-specific learning-friendly order automatically [3].
> - [4] addressed the integer addition task, discovering additional position encoding to emphasize the correspondence between digits, enhancing the length generalization.
> - [5] reported the effect of base using Integer GCD. Namely, tokenizing each integer in a particular base (e.g., 7 to 111 for base 2) has a striking impact on the learning.
>
> These are seminal works on small, focused tasks with inspiring observations. To the best of our knowledge, there is no such well-focused case study (at least explicitly) when it comes to learning medium-sensitivity. Thus, we consider this work to reserve an important position in the context of learning functions with various sensitivities. We appreciate it if the reviewer takes into account this context.
>
> [1] Power, "Grokking: Generalization Beyond Overfitting on Small Algorithmic Datasets," arXiv:2201.02177, 2022.
>
> [2] Shen et al., "Positional description matters for transformers arithmetic" arXiv:2311.14737, 2023.
>
> [3] Sato et al., "Chain of Thought in Order: Discovering Learning-Friendly Orders for Arithmetic," ICML MATH4AI Workshop, 2025
>
> [4] McLeish et al., "Transformers Can Do Arithmetic with the Right Embeddings", NeurIPS, 2024
>
> [5] Charton, "Learning the greatest common divisor: explaining transformer predictions," ICLR, 2024.

---

> ### Author Response · Authors · 2025-11-24
> **Official Comment by Authors (2/2)**
>
> ### **Q1/Q2. Discrete to continuous**
> > *“How does your medium-sensitivity framework connect to known notions of robustness or Lipschitz continuity, and can it be quantified by model capacity or gradients?”*
> > *“How does the mathematical formulation of medium-sensitivity generalize beyond the discrete symbol space?”*
>
> We appreciate these insightful questions. While the continuous case is out of our scope, we believe that these will be important in a potential future work, pursuing practical learning-based end-to-end QR code readers (i.e., images to texts).
>
> Note that our study and the literature above [1]-[5] use Transformers with tokenization and input embedding. Namely, the input is assumed to be a sequence of discrete tokens, and each token is embedded in a vector with continuous entries. Thus, working on the discrete space is a very general setup. However, in the computer vision domain, the "tokens" are already some vectors (or "tensors") with continuous values. The input embedding maps them to vectors. The continuous analysis raised by the reviewer becomes fundamental here.
>
> Connection from discrete to continuous has been widely and deeply studied in mathematics. So, the quick answer is that our definition and analysis follow it. We are not experts, but straightforwardly, the Hamming metric and difference are translated to a continuous metric (e.g., l2 norm) and differentiation. Several concepts have natural translation, but this is not always the case. For example, in discrete convex analysis, there are two categories of definitions of convexity (i.e., L-convexity and M-convexity). In continuous space, these two match. Extension from discrete to continuous is generally easier than the reverse in the robustness context.
>
> Again, while this is a bit out of scope, we appreciate the reviewer's inspiring questions. The discussion motivates us to take a step toward learning-based end-to-end QR code reading.
>
> ---
> ### **Q3. Data-dependent or non-linear noise**
> > *“Can your Reed–Solomon analysis handle non-linear or data-dependent noise, and how would that change the theoretical limits?”*
>
> We are afraid to say no. Our probabilistic analysis exploits the i.i.d. assumption, and cannot readily extend to the non-linear case and data-dependent case immediately, except for several simple cases. Such extensions are generally very hard. We need to model particular non-linearity or data dependency for the analysis, and such analysis should be done after the source of non-linearity/data dependency is identified (e.g., non-linear bit flips from physical wire transmission, lossy compression, etc.). As we discussed above, the errors from QR code detection can be an interesting source, but modeling this is out of our scope and devolves into an independent study.
>
> One possible analysis is deriving our theoretical bound for burst noise. In this case, while bit flips have some correlation, the analysis becomes very similar to the one for independent bit flips. Indeed, independent bit flips is a special case of independent burst bit flips. We are unsure if we can finish this extension during the limited rebuttal phase, but I believe that the reviewers agree that this seems quite doable. We appreciate this comment and do our best to include the extended version of our theoretical bound in the revision.
>
> ---
> ### **Q4. Failure Cases.**
> > What kinds of failure cases did you observe, and what do they reveal about the limits of medium-sensitivity learning?
>
> The examples of failure cases can be found in Tables 8 and 9; even when Transformers fail to generate complete strings, they are similar to the correct ones. This is also suggested by Figure 3, where the histogram shows the natural distributions from similarity 1 to 0.2. We don't observe any limit of medium-sensitivity learning from this result. The learning is generally successful but not completely, as we always observe in most machine learning applications.

---

### Author Response · Authors · 2025-11-24
**Global Response**

I appreciate all the reviewers for their time and effort in reviewing our manuscript. We write this short global response to make the focus of discussion clear, hopefully providing reviewers and the meta-reviewer with an overview of each reviewer's stance, the suggested major weaknesses, and our (quick) answers.

---
### **Strength.**

Most reviewers agree that our work addresses interesting and underexplored problems. The results of robustness beyond the theoretical limit and empirical analysis are also highly evaluated. The only exception is reviewer v7oJ, but we don't consider the suggested "key prior works", such as a blog post and GitHub repositories, as ruining our contributions.

**Thus, the fundamental value of this work, such as motivation, approach, and theoretical/empirical results, is accepted positively.**

---
### **Discussion Focus.**

The major weaknesses from the reviewers can be categorized into two.
1. Reviewers eM4m, qpnk are concerned that the evaluation is conducted only in the controlled setup, and also, the scale of the experiments is limited.
2. Reviewers eM4m, pS7C mentioned that computational cost is not fully analyzed and thus the practical applicability is uncertain.

We agree that these points are important for building a practical AI-based QR code reader, and the comments are valuable for future extensions. However, our goal is *not* to propose a new industrial QR decoder, but to conduct a focused case study of medium-sensitivity functions.

For this purpose, the task has to be structured enough to reveal sensitivity properties, yet simple and interpretable enough to allow controlled theoretical and experimental analysis. QR decoding is selected precisely for this reason. We prefer QR codes to artificial mathematical examples because of their familiarity; however, this familiarity directs the reviewers' focus toward practical scenarios.

We kindly ask the reviewers to
- **comprehend the case study literature of learning high-sensitivity functions (see below)** and
- **put less weight on the analysis for practical AI-based QR code readers.**


**Case study literature**

Case studies on small, restricted tasks have been evaluated as independent works with great impact in the literature of learning high-sensitivity functions. For example,

- Grokking in modular arithmetic [1]
- Order sensitivity in two-integer addition [2]
- Positional encodings for arithmetic [3]
- The choice of the encoding base number in integer GCD [4]

These are seminal works on small, focused tasks with inspiring observations. To the best of our knowledge, there is no such well-focused case study (at least explicitly) when it comes to learning medium-sensitivity.

---

### **Summary**

We consider this work to reserve an important position in the context of learning functions with various sensitivities. We appreciate it if the reviewer takes into account this context. Comprehending the scope of our work and background literature clearly is essential for the fair evaluation of our work. Note that most comments are still insightful for future extension. We appreciate them, but claim that these are not critical weaknesses of the present work.


---
[1] Power, "Grokking: Generalization Beyond Overfitting on Small Algorithmic Datasets," arXiv:2201.02177, 2022.

[2] Shen et al., "Positional description matters for transformers arithmetic" arXiv:2311.14737, 2023.

[3] McLeish et al., "Transformers Can Do Arithmetic with the Right Embeddings", NeurIPS, 2024

[4] Charton, "Learning the greatest common divisor: explaining transformer predictions," ICLR, 2024.

---

### Meta-Review · Area_Chair_NXua · 2026-01-07

**Summary:**

This paper investigates learning medium-sensitivity functions through QR code decoding with Transformers, positioning the task between low-sensitivity (image classification) and high-sensitivity (arithmetic) tasks. The authors provide theoretical analysis of error correction bounds and demonstrate empirically that Transformers achieve a high success on clean QR codes and can decode beyond theoretical limits by exploiting language structure. However, the reviewers remained not satisfied with critical concerns about missing prior work, lack of formal definitions, and insufficient scope to support broad framework claims.

**Reviewer Concerns:**

(1) Reviewer v7oJ identified specific missing prior work on neural QR decoding and extensive literature on neural error-correcting codes (Yuan, Beery, Choukroun et al.) that directly undermines novelty claims. The reviewer provided concrete references. (2) No mathematical definition is initially provided for the central concept of 'medium-sensitivity function', making claims about being 'the first case study' impossible to evaluate objectively. (3) All reviewers agree the single-task evaluation (QR decoding only) is insufficient to support the claimed framework-level contribution to medium-sensitivity learning.

**Reviewer Scores:**

The single positive reviewer (eM4m, 6) acknowledged the novel conceptual framework but raised concerns about idealized theoretical assumptions, limited experimental scale and domain, and missing computational cost analysis. The strongly negative reviewer (v7oJ, 2) provided specific evidence of missing critical citations and lack of formal definitions. After the discussion, the stance remained the same. Two reviewers (qpnk and pS7C, both 4) independently concluded the single-task evaluation is too narrow for the claimed contributions. Even if all concerns from the positive review were addressed, the fundamental issues of missing prior work, lack of formal definitions, and insufficient scope would prevent acceptance.

---

### Decision · Program_Chairs · 2026-01-26

Reject